# Where to Intervene: Action Selection in Deep Reinforcement Learning

**Wenbo Zhang**                                                                        *wenbz13@uci.edu*
*Department of Statistics*
*University of California, Irvine*

**Hengrui Cai**                                                                        *hengrc1@uci.edu*
*Department of Statistics*
*University of California, Irvine*

**Reviewed on OpenReview:** *https://openreview.net/forum?id=D3au9XkWuy*

## Abstract

Deep reinforcement learning (RL) has gained widespread adoption in recent years but faces significant challenges, particularly in unknown and complex environments. Among these, *high-dimensional action selection* stands out as a critical problem. Existing works often require a sophisticated prior design to eliminate redundancy in the action space, relying heavily on domain expert experience or involving high computational complexity, which limits their generalizability across different RL tasks. In this paper, we address these challenges by proposing a general data-driven action selection approach with model-free and computationally friendly properties. Our method not only *selects minimal sufficient actions* but also *controls the false discovery rate* via knockoff sampling. More importantly, we seamlessly integrate the action selection into deep RL methods during online training. Empirical experiments validate the established theoretical guarantees, demonstrating that our method surpasses various alternative techniques in terms of both performance in variable selection and overall achieved rewards.

## 1 Introduction

Recent advances in deep reinforcement learning (RL) have attracted significant attention, with applications spanning numerous fields such as robotics, games, healthcare, and finance (Kober et al., 2013; Kaiser et al., 2019; Kolm & Ritter, 2020; Yu et al., 2021). Despite their ability to handle sequential decision-making, the practical utility of RL methods in real-world scenarios is often limited, especially in dealing with the high-dimensional action spaces (Sunehag et al., 2015; Kaiser et al., 2019; Sakryukin et al., 2020; Xiao et al., 2020). *High-dimensional action spaces* are prevalent in "black box" systems, characterized by overloaded actionable variables that are often *abundant and redundant*. Examples include precision medicine, where numerous combinations of treatments and dosages are possible (see e.g., Johnson et al., 2016; Liu et al., 2017; Cai et al., 2023a); neuroscience, which involves various stimulation points and intensities (see e.g., Gershman et al., 2009); and robotics, particularly in muscle-driven robot control, where coordination of numerous muscles is required (see e.g., Schumacher et al., 2023). Nevertheless, these high-dimensional action spaces often contain many actions that are either ineffective or have a negligible impact on states and rewards. Training RL models on the entire action space can result in substantial inefficiencies in both computation and data collection.

To handle high dimensionality, a promising approach is to employ automatic dimension reduction techniques to select only *the essential minimum action set* necessary for effectively learning the environment and optimizing the policy based on the subspace. Having such a minimal yet sufficient action space can significantly enhance learning efficiency, as agents can thoroughly explore a more concise set of actions (Zahavy et al.,

2018; Kanervisto et al., 2020; Jain et al., 2020; Zhou et al., 2024). Moreover, a smaller action space can reduce computational complexity, a notable benefit in deep RL, where neural networks are used for function approximation (Sun et al., 2011; Sadamoto et al., 2020). In practical scenarios, eliminating superfluous actions saves the cost of extensive measurement equipment and thus allows a more comprehensive exploration of available actions. Yet, existing works often require a sophisticated prior design to eliminate redundancy in the action space (e.g., Synnaeve et al., 2019; Jiang et al., 2019; Farquhar et al., 2020; Luo et al., 2023), relying heavily on domain expert experience or involving high computational complexity, limiting their generalizability across different RL tasks.

In this paper, we propose a general data-driven action selection approach to identify the minimum sufficient actions in the high-dimensional action space. To handle the complex environments often seen in deep RL, we develop a novel variable selection approach called knockoff sampling (KS) for online RL, with theoretical guarantees of *false discovery rate control*, inspired by the model-free knockoff method (Candes et al., 2018). The effectiveness of this action selection method is demonstrated in Fig. 1. A proximal policy optimization (PPO) algorithm (Schulman et al., 2017) enhanced with variable selection outperforms its counterpart without selection and achieves performance comparable to that of PPO trained with the pre-known true minimal sufficient action. To remain computationally friendly, we design an adaptive strategy with a simple mask operation that seamlessly integrates this action selection method into deep RL methods during online training. Our method does not reduce the dimensionality of the action space itself but rather focuses on identifying redundant actions and mitigating their influence during policy training to enhance sample efficiency and significantly accelerate learning with implementation and interpretation advantages.

Our main **contributions** are fourfold:

• Conceptually, this work pioneers exploring high-dimensional action selection in online RL. We formally define *the sufficient action set* as encompassing all influential actions and *the minimal sufficient action set* as containing the smallest number of actions necessary for effective decision-making.

• Methodologically, our method *bypasses the common challenge of creating accurate knockoff features in model-free knockoffs*. We use the established distribution of actions from the current policy network in online RL to resample action values, producing exact knockoff features.

• Algorithm-wise, to *flexibly integrate arbitrary variable selection into deep RL* and eliminate the need to initialize a new RL model after the selection, we design a binary hard mask approach based on the indices of selected actions. This efficiently neutralizes the influence of non-chosen actions.

• Theoretically, to *address the issues of highly dependent data* in online RL, we couple our KS method with sample splitting and majority vote; under commonly imposed conditions, we theoretically show our method *consistently* identifies the minimal sufficient action set with false discovery rate control.

## 1.1 Related Works

**Deep reinforcement learning** has made significant breakthroughs in complex sequential decision-making across various tasks (Mnih et al., 2013; Silver et al., 2016; Schulman et al., 2017; Haarnoja et al., 2018; Cai et al., 2021). Yet, several considerable obstacles exist when dealing with high-dimensional spaces using deep RL. In terms of *high dimensional state space*, the state abstraction (Misra et al., 2020; Pavse & Hanna, 2023) has been studied to learn a mapping from the original state space to a much smaller abstract space to preserve the original Markov decision process. Yet, these methods, such as bisimulation can be computationally expensive and challenging when the state space is very large or has complex dynamics (Ruan et al., 2015). Tied to our topic, it is *hard to utilize such abstraction-based methods to implement transformed actions*. This redirects us to *variable selection* on the redundant state space (see e.g., Kroon & Whiteson, 2009; Guo & Brunskill, 2017). Recently, Hao et al. (2021) combined LASSO with fitted Q-iteration to reduce states; following this context, Ma et al. (2023) employs the knockoff method for state selection but with discrete action spaces. However, all these works focus on the high dimensional state space in offline data, while our method aims to extract sufficient and necessary actions during online learning.

**For RL with the high-dimensional action space**, especially for continuous actions, some studies (Synnaeve et al., 2019; Farquhar et al., 2020) transformed the continuous control problem into the combinatorial action

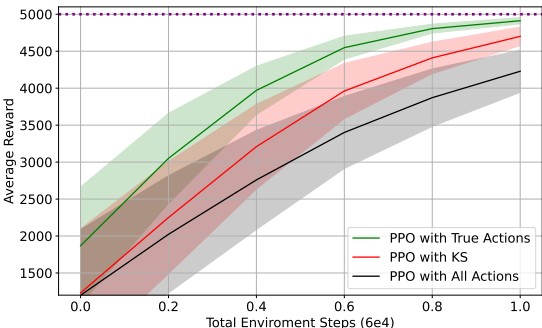

Figure 1: Average rewards under three proximal policy optimization (PPO) methods in a synthetic environment with 54 actions (among which only 4 actions influence states and rewards). The green line refers to the PPO trained based on the true influential actions, the red line refers to the PPO with the estimated minimal sufficient actions by the proposed variable selection (KS), the black line represents the PPO with the entire redundant action space, and the dashed line is the optimal reward. The red line outperforms the black line, indicating *the effectiveness of the variable selection* step.

problem, by discretizing large action spaces into smaller subspaces. However, this transformation can lead to a significant loss of precision and hence produce suboptimal solutions (Lee et al., 2018; Tan et al., 2019; Li et al., 2023). Other works (see e.g., Jiang et al., 2019; Luo et al., 2023) focused on muscle control tasks and used architectures reducing the action dimensionality before deploying RL methods. One recent study by Schumacher et al. (2023) combined differential extrinsic plasticity with RL to control high-dimensional large systems. Yet, all these works require specialized data collection, known joint ranges of actions, forced dynamics, or desired behaviors of policies, before implementing RL. In contrast, our method is entirely *data-driven without prior knowledge of environments* and thus can be generalized to tasks beyond muscle control. Some studies (Zahavy et al., 2018; Zhong et al., 2024) have also explored eliminating actions; however, their approaches are limited to discrete action spaces and either require explicit elimination signals provided by the environment or fit an inverse dynamics model of the environment.

**Variable selection**, also known as feature selection, is a critical process to choose the most relevant variables representing the target outcome of interest, enhancing both model performance and interpretation. Over the past few decades, many well-known methods have been established, ranging from classical LASSO, Fisher score, and kernel dimension reduction (Tibshirani, 1996; Gu et al., 2012; Chen et al., 2017; Zhou et al., 2021), towards deep learning (Liang et al., 2018; Balın et al., 2019; Lee et al., 2021; Cai et al., 2023b; Zhang et al., 2023). Yet, these works either *suffer from model-based constraints or lack theoretical guarantees.* **The model-X knockoff method** proposed by Candes et al. (2018) aims to achieve both goals via a general variable selection framework for black-box algorithms with guarantees of false discovery rate control. Due to its model-agnostic nature, the knockoff method has been extended to complement a wide range of variable selection approaches (Sesia et al., 2017; Ma et al., 2021; Liu et al., 2022). The main price or central challenge within the knockoff method lies in *the generation of faithful knockoff features.* Existing techniques either use model-specific methods (see e.g., Sesia et al., 2017; Liu & Zheng, 2018) that assume the underlying covariate distribution, or model-free approaches (see e.g., Jordon et al., 2018; Romano et al., 2020) that utilize deep generative models to obtain knockoffs without further assumptions on feature distribution. Owing to the blessing of online RL, our method bypasses this challenge through the known joint distribution of actions represented by the ongoing policy network, and thus can easily resample the action values to create exact knockoff features.

## 2 Problem Setup

### 2.1 Notations

Consider a Markov Decision Process (MDP) characterized by the tuple $(\mathcal{S}, \mathcal{A}, p, r, \gamma)$, in which both the state space $\mathcal{S}$ and the action space $\mathcal{A}$ are continuous. The state transition probability, denoted as $p : \mathcal{S} \times \mathcal{S} \times \mathcal{A} \rightarrow$

$[0, \infty)$, is an unknown probability density function that determines the likelihood of transitioning to a next state $\mathbf{s}_{t+1} \in \mathcal{S}$, given the current state $\mathbf{s}_t \in \mathcal{S}$ and the action $\mathbf{a}_t \in \mathcal{A}$. The environment provides a reward, bounded within $[r_{\min}, r_{\max}]$, for each transition, expressed as $r : \mathcal{S} \times \mathcal{A} \to [r_{\min}, r_{\max}]$. The discount factor, represented by $\gamma \in (0, 1)$, influences the weighting of future rewards. We denote a generic tuple consisting of the current state, action, reward, and subsequent state as $(\mathbf{S}_t, \mathbf{A}_t, R_t, \mathbf{S}_{t+1})$. The Markovian property of MDP is that given the current state $\mathbf{S}_t$ and action $\mathbf{A}_t$, the current $R_t$ and the next state $\mathbf{S}_{t+1}$ are conditionally independent of the past trajectory history. Consider $\mathbf{A}_t \in \mathbb{R}^p$ where $p$ is very large indicating a *high dimensional action space*. We utilize $\rho_\pi(\mathbf{s}_t)$ and $\rho_\pi(\mathbf{s}_t, \mathbf{a}_t)$ to denote the state and state-action marginal distributions, respectively, of the trajectory distribution generated by a policy $\pi(\mathbf{a}_t \mid \mathbf{s}_t)$. The notation $J(\pi)$ is used to represent the expected discounted reward under this policy: $J(\pi) = \sum_t \mathbb{E}_{(\mathbf{s}_t, \mathbf{a}_t) \sim \rho_\pi} [\gamma^t r(\mathbf{s}_t, \mathbf{a}_t)]$. The goal of RL is to maximize the expected sum of discounted rewards above. This can be extended to a more general maximum entropy objective with the expected entropy of the policy over $\rho_\pi(\mathbf{s}_t)$.

## 2.2 Minimal Sufficient Action Set in Online RL

To address the high-dimensional action space, we propose to utilize variable selection instead of representation for practical usefulness. To achieve this goal, we first formally define the minimal sufficient action set. Denote the subvector of $\mathbf{A}_t$ indexed by components in $G$ as $\mathbf{A}_{t,G}$ with an index set $G \subseteq \{1, 2, \ldots, p\}$. Let $G^c = \{1, \ldots, p\} \backslash G$ be the complement of $G$.

**Definition 2.1. (Sufficient Action Set)** We say $G$ is the sufficient action (index) set in an MDP if

$$R_t \perp \mathbf{A}_{t,G^c} \mid \mathbf{S}_t, \mathbf{A}_{t,G}, \qquad \mathbf{S}_{t+1} \perp \mathbf{A}_{t,G^c} \mid \mathbf{S}_t, \mathbf{A}_{t,G}, \qquad \text{for all } t \geq 0.$$

The sufficient action set can be seen as a sufficient conditional set to achieve past and future independence. The sufficient action set may not be unique.

**Definition 2.2. (Minimal Sufficient Action Set)** We say $G$ is the minimal sufficient action set in an MDP if it has the smallest cardinality among all sufficient action sets.

Unlike the sufficient action set, there is only one unique minimal sufficient action set to achieve conditional independence if there are no identical action variables in the environment. We also call $G^c$ the redundant set when $G$ is the minimal sufficient action (index) set. Here, to achieve such a minimal sufficient action set, one should also require the states $\mathbf{S}_t$ to be the sufficient states, so there is no useless state (Ma et al., 2023) to introduce related redundant actions that possibly lead to ineffective exploration or data inefficiency. Without loss of generality, we assume sufficient states throughout this paper and focus on eliminating the influence of redundant actions in a high-dimensional action space. Our goal is to identify the minimal sufficient action set for online deep reinforcement learning to improve exploration.

## 2.3 Preliminary: Knockoff Variable Selection

Without making additional assumptions on the dependence among variables, in this work, we utilize the model-X knockoffs (Candes et al., 2018) for flexible variable selection, which ensures finite-sample control of the false discovery rate (FDR). We first briefly review *the model-X knockoffs (Candes et al., 2018) in the supervised regression setting with independent samples*, which will be leveraged later *as the base variable selector of our proposed method for dependent data in the online RL setting*.

Suppose we have $n$ i.i.d. samples from a population, each of the form $(X, Y)$, where $X = (X_1, \ldots, X_p) \in \mathbb{R}^p$ and the outcome $Y \in \mathbb{R}$. We further denote $\boldsymbol{Y}$ as an $n$-dimensional response vector and $\boldsymbol{X}$ as an $n \times p$ matrix of covariates by aggregating $n$ samples. The variable selection problem stems from the fact that, in many real-world scenarios, the outcome variable $Y$ is influenced by only a small subset of the predictors $X$. Formally, the goal is to identify a subset of indices $G \subset \{1, \ldots, p\}$, with $|G| < p$, such that the conditional distribution $F_{Y|X}$ depends only on the variables $\{X_j\}_{j \in G}$, and $Y$ is conditionally independent of the remaining variables given this subset. That is,

$$Y \perp X_j \mid \{X_k\}_{k \in G} \quad \text{for all } j \notin G.$$

The variable selection is looking for the Markov blanket $G$, i.e., the "smallest" subset $G$ such that conditionally on $\{X_j\}_{j \in G}$, $Y$ is independent of all other variables. To ensure the uniqueness of relevant variables in the Markov blanket, pairwise independence is introduced as follows.

**Definition 2.3.** A variable $X_j$ is said to be "null" if and only if $Y$ is independent of $X_j$ conditionally on the other variables $X_{-j} = \{X_1, \ldots X_p\} \backslash \{X_j\}$, otherwise said to be "nonnull" or relevant. The set of null variables is denoted by $\mathcal{H}_0 \subset \{1, \ldots p\}$.

For a selected subset $\widehat{G}$ of the covariates constructed from data and some pre-specified level $q \in (0, 1)$, the false discovery rate (FDR) and modified FDR (mFDR) associated with $\widehat{G}$ are formally defined as FDR := $\mathbb{E}\{|\widehat{G} \bigcap \mathcal{H}_0|/\max(1, |\widehat{G}|)\}$ and mFDR := $\mathbb{E}\{|\widehat{G} \bigcap \mathcal{H}_0|/(1/q + |\widehat{G}|)\}$, respectively, where $|\cdot|$ denotes the cardinality of a set. Here, FDR is the expected proportion of falsely selected variables among the selected set, and mFDR offers a less conservative measurement by adjusting the denominator. Knockoff variable selection aims to discover as many relevant (conditionally dependent) variables as possible while keeping the FDR under control.

Towards this goal, the model-X knockoff generates an $n \times p$ matrix $\widetilde{\boldsymbol{X}} = (\widetilde{\boldsymbol{x}}_1, \cdots, \widetilde{\boldsymbol{x}}_p)$ as *knockoff features* that have the similar properties as the collected covariates. This matrix is constructed by the joint distribution of $\boldsymbol{X}$ and satisfies:

$$\widetilde{\boldsymbol{X}} \perp \boldsymbol{Y} \mid \boldsymbol{X} \quad \text{and} \quad (\boldsymbol{X}, \widetilde{\boldsymbol{X}})_{\text{swap}(\Omega)} \stackrel{d}{=} (\boldsymbol{X}, \widetilde{\boldsymbol{X}}), \tag{1}$$

for each subset $\Omega$ within the set $\{1, \cdots, p\}$, where swap($\Omega$) indicates the operation of swapping such that for each $j \in \Omega$, the $j$-th and $(j+p)$-th columns are interchanged. Here, the swap is performed between corresponding coordinates of $\boldsymbol{X}$ and $\widetilde{\boldsymbol{X}}$, and we must ensure that the knockoffs are constructed such that the joint distribution remains invariant under coordinate-wise swaps. The notation $\stackrel{d}{=}$ signifies equality in distribution. After obtaining knockoff features, let $\widetilde{\mathcal{D}} = \{\boldsymbol{X}, \widetilde{\boldsymbol{X}}, \boldsymbol{Y}\}$ denote an augmented dataset and we can calculate the feature importance scores $Z_j$ and $\widetilde{Z}_j$ for each variable $\boldsymbol{x}_j$ and its corresponding knockoff $\widetilde{\boldsymbol{x}}_j$ based on any regression or machine learning methods like lasso or random forest. Define the function $f : \mathbb{R}^2 \to \mathbb{R}$ as an anti-symmetric function, meaning that $f(u, v) = -f(v, u)$ for all $u, v \in \mathbb{R}^2$, e.g., $f(u, v) = u - v$. Set knockoff statistics $\mathbf{W} = (W_1, \ldots, W_p)^\top$ where $W_j = f(Z_j, \widetilde{Z}_j)$ in such a way that higher values of $W_j$ indicate stronger evidence of the significance of $\boldsymbol{x}_j$ being influential covariate. The $j$-th variable is selected if its corresponding $W_j$ is at least a certain threshold $\tau_\alpha$ when the target FDR level is $\alpha$. Then the set of chosen variables can be represented as $\widehat{\mathcal{I}} = \{j : W_j \geq \tau_\alpha\}$, where

$$\tau_\alpha = \min \left\{ \tau > 0 : \frac{\# \{j \in [p] : W_j \leq -\tau\}}{\# \{j \in [p] : W_j \geq \tau\}} \leq \alpha \right\}. \tag{2}$$

The knockoff procedure controls the mFDR at a pre-specified level $\alpha$, ensuring that the approximated expected proportion of false positives among the selected variables does not exceed $\alpha$. A more conservative variant, known as Knockoffs+, can be used to provide guaranteed control of the FDR. More details can be found in (Candes et al., 2018).

## 3 Online Deep RL with Variable Selection

To identify the minimal sufficient action set in online deep RL, we integrate the action selection into RL to find truly influential actions during the training process. Its advantages are manifold. Firstly, its model-agnostic nature ensures compatibility across various RL architectures and algorithms. Moreover, its data-driven characteristic allows for straightforward application across diverse scenarios, thereby increasing practical utility. Crucially, the action selection boosts the explainability and reliability of RL systems by clearly delineating actions that contribute to model performance. In the following, we first introduce an action-selected exploration strategy for online deep RL in Section 3.1, followed by the model-free knockoff-sampling method for action selection in Section 3.2.

### 3.1 Action-Selected Exploration Algorithm

We propose an innovative action-selected exploration for deep RL. Suppose at a predefined time step $t = T_{vs}$, a set of actions $\widehat{G}$ is identified from the buffered data, where the cardinality of $\widehat{G}$ $(|\widehat{G}|)$ is $d$, with $d \leq p$ indicting a size of selected actions. A critical challenge arises in leveraging the insights gained from action selection for updating the deep RL models. The conventional approach of constructing an entirely new model based on the selected actions is not only time-consuming but also inefficient, particularly in dynamic, non-stationary environments where the requisite action sets are subject to frequent changes. Although our study primarily focuses on stationary environments, the inefficiency of model reinitialization post-selection remains a notable concern.

To seamlessly and efficiently integrate action selection results into deep RL, we propose to mask the non-selected actions and remove their influence once a hard mask is constructed, and thus *is flexible to integrate with arbitrary variable selection method*. Specifically, in continuous control tasks, deep RL algorithms utilize a policy network $\pi_\theta$ to sample a certain action $\mathbf{a}$ given current state $\mathbf{s}$, namely $\mathbf{a} \sim \pi_\theta (\cdot \mid \mathbf{s})$. Here, we use the Gaussian policy as an illustrative example, but it can be flexibly generalized to other distributions. Assume the policy network is parameterized by a multivariate Gaussian with the diagonal covariance matrix as:

$$\mathbf{a} \sim \mathcal{N} \left( \mu\left(\mathbf{s}\right), \mathrm{diag}\left(\sigma\left(\mathbf{s}\right)\right)^2 \right),$$

where $\mu$ and $\sigma$ are parameterized functions to output mean and standard deviations. Each time we obtain an action from the policy network. The updates of the policy network $\pi_\theta$ and the action-value function $Q_\phi$ usually involve sampled actions $\mathbf{a}_t$ and $\log \pi_\theta (\mathbf{a}_t \mid \mathbf{s}_t)$, which is the log density of sampled actions. Our strategy is to use *a binary mask* to set them to a certain constant value during the forward pass, and it will block the gradient when doing backpropagation and also remove influence when fitting a function. Given a selected action set $\widehat{G}$, we focus on integrating this selection into the model components $Q_\phi(\mathbf{a}, \mathbf{s})$ and $\pi_\theta(\mathbf{a} \mid \mathbf{s})$. To facilitate this, we define a selection vector $\mathbf{m} = (m_1, \cdots, m_p) \in \{0, 1\}^p$, where $m_i = 1$ if $i \in \widehat{G}$ and 0 otherwise. This vector enables the application of a selection mask to both the $Q_\phi$ and $\pi_\theta$ as follows.

For $Q_\phi$, we use the hard mask to remove the influence of non-selected actions during $Q$ function fitting,

$$Q_\phi^m(\mathbf{a}, \mathbf{s}) = Q_\phi(\mathbf{m} \odot \mathbf{a}, \mathbf{s}), \tag{3}$$

where $\odot$ is the element-wise product. The adoption of action selection can reduce bias in the $Q$ function fitting when sufficient action is correctly identified. It can also reduce variance by decreasing the complexity of the hypothesis space of the $Q$ function.

For $\pi_\theta$, considering the necessity of updating the policy network via policy gradient, we integrate a hard mask into the logarithm of the policy probability. The modified log probability is formulated as

$$\log \pi_\theta^m(\mathbf{a} \mid \mathbf{s}) = \mathbf{m} \cdot (\log \pi_\theta(a_1 \mid \mathbf{s}), \ldots, \log \pi_\theta(a_p \mid \mathbf{s})), \tag{4}$$

where $\cdot$ is the dot product. This masking of the log probability helps mitigate the likelihood of encountering extremely high entropy values, thereby facilitating a more stable and efficient training process. We demonstrate the integration of action selection into deep RL as detailed in Algorithm 1.

**Remark 3.1.** Here, we focus on the case where actions are parameterized as diagonal Gaussian which are conditionally independent given states. However, our method can be easily extended to the correlated actions, with details provided in Appendix F.

**Remark 3.2.** In scenarios where the algorithm exclusively employs the state-value function $V_\phi(\mathbf{s})$, the use of the mask operation is unnecessary. Our empirical studies suggest that, even without masking, the model maintains robust performance. This implies that updates to the policy network may hold greater significance than those to the critic in certain contexts.

### 3.2 Knockoff-Sampling for Action Selection

Despite the large volume of variable selection (VS) methods (see e.g., Tibshirani, 1996; Gu et al., 2012; Chen et al., 2017; Liang et al., 2018; Balın et al., 2019; Lee et al., 2021), these works either *suffer from*

---

**Algorithm 1** Action-Selected Exploration in Reinforcement Learning

---

**Require:** FDR rate $\alpha$, majority voting ratio $\Gamma$, max steps $T$, variable selection step $T_{vs}$
  **Begin:** Initialize the selection set $\widehat{G} = \{\}$, policy $\pi_\theta$, value function parameter $\phi$, augmented replay buffer $\mathcal{D}$
  **while** steps smaller than $T$ **do**
    Sample $\mathbf{a}_t \sim \pi_\theta \left( \cdot \mid \mathbf{s}_t \right)$
    Sample knockoff copy $\widetilde{\mathbf{a}}_t \sim \pi_\theta \left( \cdot \mid \mathbf{s}_t \right)$
    $\mathbf{s}_{t+1} \sim \mathrm{Env} \left( \mathbf{a}_t, \mathbf{s}_t \right)$
    $\mathcal{D} \leftarrow \mathcal{D} \cup \{\mathbf{s}_t, \mathbf{a}_t, \widetilde{\mathbf{a}}_t, r_t, \mathbf{s}_{t+1}\}$
    **if** $t = T_{vs}$ **then**
      Utilize a variable selection algorithm (optional: Knockoff-Sampling in Algorithm 2) on $\mathcal{D}$ to obtain the estimated minimal sufficient action set $\widehat{G}$
      Generate a mask $\mathbf{m}$ based on $\widehat{G}$ to prune RL networks based on equation 3 and equation 4
    **end if**
    **if** it's time to update **then**
      update $\phi$ and $\theta$ based on the specific RL algorithm used
    **end if**
  **end while**

---

**Algorithm 2** Knockoff-Sampling Variable Selection

---

**Require:** FDR rate $\alpha$, majority voting ratio $\Gamma$, data buffer $\mathcal{D} = \{(\mathbf{s}_t, \mathbf{a}_t, \widetilde{\mathbf{a}}_t, r_t, \mathbf{s}_{t+1})\}_{t=1}^{T_{vs}}$
  Split $\mathcal{D}$ into non-overlapping sets $\{\mathcal{D}_k\}_{k=1}^K$ and let $\mathbf{y}_t = (r_t, \mathbf{s}_{t+1})$ as the response vector
  **for** $k = 1, \ldots K$ **do**
    **for** $i$-th dimension in $\{\mathbf{y}_t\}_{t=1}^{T_{vs}}$ **do**
      Apply a machine learning algorithm to all $(\mathbf{s}_t, \mathbf{a}_t, \widetilde{\mathbf{a}}_t, \mathbf{y}_t) \in \mathcal{D}_k$ to construct feature importance statistics $Z_{j,i}$ and $\widetilde{Z}_{j,i}$ for the $j$-th action and its knockoff copy, respectively, for each $j \in [p]$.
    **end for**
    **for** each $j \in [p]$ **do**
      Set    $Z_j = \max_i Z_{j,i}, \quad \widetilde{Z}_j = \max_i \widetilde{Z}_{j,i}, \quad$ and $\quad W_j = f\left( Z_j, \widetilde{Z}_j \right)$
    **end for**
    Utilize the threshold $\tau_\alpha$ defined in equation 2, and get $\widehat{G}_k = \{j \in [p] : W_j \geq \tau_\alpha\}$
  **end for**
  **return** $\widehat{G} := \left\{ j \in \{1, \ldots, p\} : \sum_{k=1}^K \mathbb{I}\left( j \in \widehat{G}_k \right) \geq K\Gamma \right\}$

---

*model-based constraints or lack theoretical guarantees.* The traditional VS often identifies unimportant actions, leading to a high false discovery rate and further causing performance degeneration, as shown in Fig. 2. To provide a general action selection approach for deep RL with false discovery rate control, we propose a novel knockoff-sampling (KS) method that handles dependent data in the online setting with a model-agnostic nature as follows.

Suppose now we have a data buffer with the size $M$, collected from $N$ trajectories where each trajectory has length $T_j$ for $j = 1, \ldots N$ and $\sum_{j=1}^N T_j = M$. Each time we obtain an action from the policy network, we also resample a knockoff copy conditional on the same state $\widetilde{\mathbf{a}}_t \sim \pi_\theta \left( \cdot \mid \mathbf{s}_t \right)$, and append it to the buffer. The transition tuples thus is redefined as $(\mathbf{S}_t, \mathbf{A}_t, \widetilde{\mathbf{A}}_t, R_t, \mathbf{S}_{t+1})$. Note that steps within each trajectory are temporally dependent. To *address the issues of highly dependent data* in online RL, we couple our method with sample splitting and majority vote following Ma et al. (2023). The proposed KS method consists of three steps as summarized in Algorithm 2: 1. Sample Splitting; 2. Knockoff-Sampling Variable Selection; 3. Majority Vote. We detail each step below.

1. **Sample Splitting:** We first split all transition tuples $(\mathbf{S}_t, \mathbf{A}_t, \widetilde{\mathbf{A}}_t, R_t, \mathbf{S}_{t+1})$ into $K$ non-overlapping sub-datasets. This process results in a segmentation of the dataset $\mathcal{D}$ into distinct subsets $\mathcal{D}_k$ for $k \in [K]$. We combine response variables and denote $\mathbf{Y}_t = (R_t, \mathbf{S}_{t+1})$ to simplify the notation, based on the target

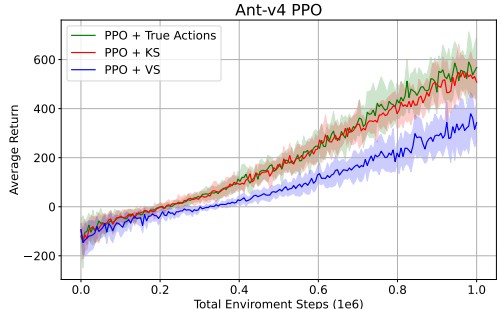
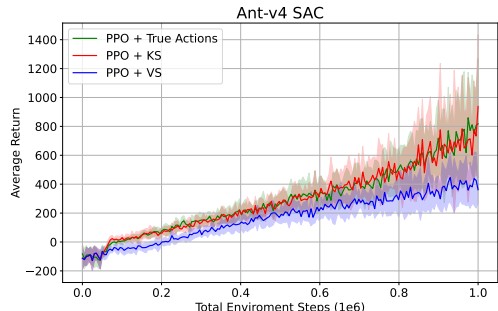

Figure 2: Learning curves in the Ant-v4 environment reveal that the Knockoff Sampling (KS) method outperforms the traditional Variable Selection (VS) method. When implemented with either the Proximal Policy Optimization (PPO) or Soft-Actor-Critic (SAC) algorithm, KS achieves performance comparable to that of the true actions by automatically setting optimal thresholds to filter out redundant actions, whereas VS often selects useless actions, leading to a high false discovery rate.

outcomes in Definition 2.1. Here, each sequence $(\mathbf{S}_t, \mathbf{A}_t, \widetilde{\mathbf{A}}_t, \mathbf{Y}_t)$ is assigned to $\mathcal{D}_k$ if $t \mod K = k - 1$. Subsequent to this division, any two sequences located within the same subset $\mathcal{D}_k$ either originate from the same trajectory with a temporal separation of no less than $K$ or stem from different trajectories. If the system adheres to $\beta$-mixing conditions (Bradley, 2005), then a careful selection (Berbee, 1987) can allow us to assert that transition sequences within each subset $\mathcal{D}_k$ are approximately independent.

2. **Knockoff-Sampling Variable Selection:** For each data subset $\mathcal{D}_k$, we select a minimal sufficient action set using the model-X knockoffs as the base selector. Unlike the knockoff method detailed in Section 2.3 that either constructs knockoff features based on second-order machines or estimates the full distribution, we directly sample a knockoff copy of actions from the policy network, i.e., $\widetilde{\mathbf{a}}_t \sim \pi_\theta (\cdot \mid \mathbf{s}_t)$. This helps us to *bypass the common challenge of creating accurate knockoff features in model-free knockoffs.* We theoretically validate that the sampled knockoffs in online RL meet the swapping property equation 1 in Section 4. For every single dimension $i$ of the outcome vector $\mathbf{Y}_t = (R_t, \mathbf{S}_{t+1})$, we use a general machine learning method (e.g., LASSO, random forest, neural networks) to provide variable importance scores $Z_{j,i}$ and $\widetilde{Z}_{j,i}$ for the $j$-th dimension of actions and its knockoff copy, respectively. By the maximum score $Z_j = \max_i Z_{j,i}$ and $\widetilde{Z}_j = \max_i \widetilde{Z}_{j,i}$, the selected action set $\widehat{G}_k$ is then obtained following the same procedure in Section 2.3 after computing knockoff statistics $\mathbf{W}$.

3. **Majority Vote:** To combine the results on the whole $K$ folds, we calculate the frequency of subsets where the $j$-th action is chosen, i.e., $\widehat{p}_j = \sum_{k=1}^{K} \mathbb{I}(j \in \widehat{G}_k)/K$, and establish the ultimate selection of actions $\widehat{G} = \{j : \widehat{p}_j \geq \Gamma\}$, with $\Gamma$ being a predetermined cutoff between 0 and 1.

## 4 Theoretical Results

Without loss of generality, we assume that the data buffer $\mathcal{D}$ consists of $N$ i.i.d. finite-horizon trajectories, each of length $T$, which can be summarized as $NT$ transition tuples. We first define two properties to establish theoretical results.

**Definition 4.1.** (**Flip Sign Property for Augmented Data**) Knockoff statistics $\mathbf{W} = (W_1, \ldots, W_p)^\top$ satisfies property on the augmented data matrix $\mathcal{D}_k = \left[\mathbf{A}_k, \tilde{\mathbf{A}}_k, \mathbf{S}_k, \mathbf{Y}_k\right]$ if for any $j \in [p]$ and $\Omega \subset [p]$,

$$W_i\left(\left[\mathbf{A}_k, \tilde{\mathbf{A}}_k\right]_{\mathrm{swap}(\Omega)}, \mathbf{S}_k, \mathbf{Y}_k\right) = \begin{cases} -W_i\left(\left[\mathbf{A}_k, \tilde{\mathbf{A}}_k\right], \mathbf{S}_k, \mathbf{Y}_k\right), & \text{if } j \in \Omega, \\ W_i\left(\left[\mathbf{A}_k, \tilde{\mathbf{A}}_k\right], \mathbf{S}_k, \mathbf{Y}_k\right), & \text{otherwise}, \end{cases}$$

where $\mathbf{A}_k, \tilde{\mathbf{A}}_k \in \mathbb{R}^{(NT/K) \times p}$ denote the matrices of the actions and their knockoffs, $\mathbf{S}_k \in \mathbb{R}^{(NT/K) \times d}$ denote the matrice of state, $\mathbf{Y}_k \in \mathbb{R}^{(NT/K) \times d+1}$ denotes the response matrix, and $\left[\mathbf{A}_k, \tilde{\mathbf{A}}_k\right]_{\mathrm{swap}(\Omega)}$ is obtained by swapping all $j$-th columns in $\mathbf{A}_k, \tilde{\mathbf{A}}_k$ for $j \in \Omega$.

The above flip sign property is a common property that needs to be satisfied in knockoff-type methods. We show that our method automatically satisfies this property in Lemma I.3 of the Appendix.

We now define the $\beta$-mixing coefficient, which quantifies the strength of dependence between observations separated by a time lag in a stochastic process.

**Definition 4.2.** ($\beta$-mixing Coefficient, Bradley (2005)). For a sequence of random variables $\{X_t\}$, define its $\beta$-mixing coefficient as

$$\beta(i) := \sup_{m \in \mathbb{Z}} \beta\left(\mathcal{F}_{-\infty}^m, \mathcal{F}_{i+m}^\infty\right)$$

Using this definition, we now introduce the concept of exponential $\beta$-mixing, which describes processes where the dependence decays at an exponential rate.

**Definition 4.3.** (**Stationarity and Exponential $\beta$-Mixing**) The process $\{(\mathbf{S}_t, \mathbf{A}_t, R_t)\}_{t \geq 0}$ is stationary and exponentially $\beta$-mixing if its $\beta$-mixing coefficient at time lag $k$ is of the order $\rho^k$ for some $0 < \rho < 1$

This exponential $\beta$-mixing condition has been assumed in the RL literature (see e.g., Antos et al., 2008; Dai et al., 2018) to derive the theoretical results for the dependent data. Such a condition quantifies the decay in dependence as the future moves farther from the past to achieve the dependence of the future on the past. Based on the above definitions, we establish the following false discovery control results of our method.

**Theorem 4.4.** *Set the number of sample splits $K = k_0 \log(NT)$ for some $k_0 > -\log^{-1} \rho$ where $\rho$ is defined in Definition 4.3. Assume that the following assumption hold: the process $\{(\mathbf{S}_t, \mathbf{A}_t, R_t)\}_{t \geq 0}$ is stationary and exponentially $\beta$-mixing.*

*Then $\widehat{G}_k$ obtained by Algorithm 2 with the standard knockoffs controls the modified FDR (mFDR),*

$$mFDR \leq \alpha + O\left\{K^{-1}(NT)^{-c}\right\},$$

*where the constant $c = -k_0 \log(\rho) - 1 > 0$.*

The proof can be mainly divided into two parts. Firstly, we show that valid mFDR control can be achieved when data are independent. Then, for dependent data satisfying the $\beta$-mixing condition, the upper bound can be relaxed to account for the cost of dependence, which vanishes as the number of samples in $\mathcal{D}_k$ approaches infinity. Finally, a combination of the two would provide the final upper bound on mFDR control. The detailed proof is in Appendix I. This theorem provides a theoretical guarantee for controlling the modified false discovery rate by our proposed method at a pre-specified level $\alpha$ with a small order term that goes to zero as the sample size goes to infinity. In reinforcement learning, this ensures the identification of a minimal sufficient set of actions with a controlled error tolerance, which is essential for efficient learning. Our method selects and only utilizes those relevant actions, avoiding irrelevant ones that could degrade policy performance and lead to unnecessary costs.

## 5 Experiments

**Experiment Setup** We aim to answer whether the variable selection is helpful for deep RL training when the action dimension is high and redundant. We conduct experiments on standard locomotion tasks in MuJoCo (Todorov et al., 2012) and treatment allocation tasks calibrated from electronic health records (EHR), the MIMIC-III dataset (Johnson et al., 2016). The environment details are in Table C.1 of the Appendix. Here, we focus on two representative actor-critic algorithms, Proximal Policy Optimization (PPO) (Schulman et al., 2017) and Soft-Actor-Critic (SAC) (Haarnoja et al., 2018). We adopt the implementation from Open AI Spinning up Framework (Achiam, 2018). For SAC, the implementation involves fitting both $Q_\phi$ and $\pi_\theta$, and we use a mask on both components. For PPO, it fits $V_\theta$ and $\pi_\theta$, hence we only combine the mask with $\pi_\theta$. Tables B.1 and B.2 summarize the hyperparameters we used. We set the FDR rate $\alpha = 0.1$ and voting ratio $\Gamma = 0.5$ in all settings. All the experiments are conducted on the server with $4\times$ NVIDIA RTX A6000 GPU.

**Semi-synthetic MuJoCo Environments** We chose three tasks: Ant, HalfCheetah, and Hopper. To increase the dimension of action space, we artificially add extra $p$ actions to the raw action space and consider two scenarios, $p = 20$ and $50$. For each setting, we run experiments over $2 \times 10^5$ and $10^6$ steps for SAC and

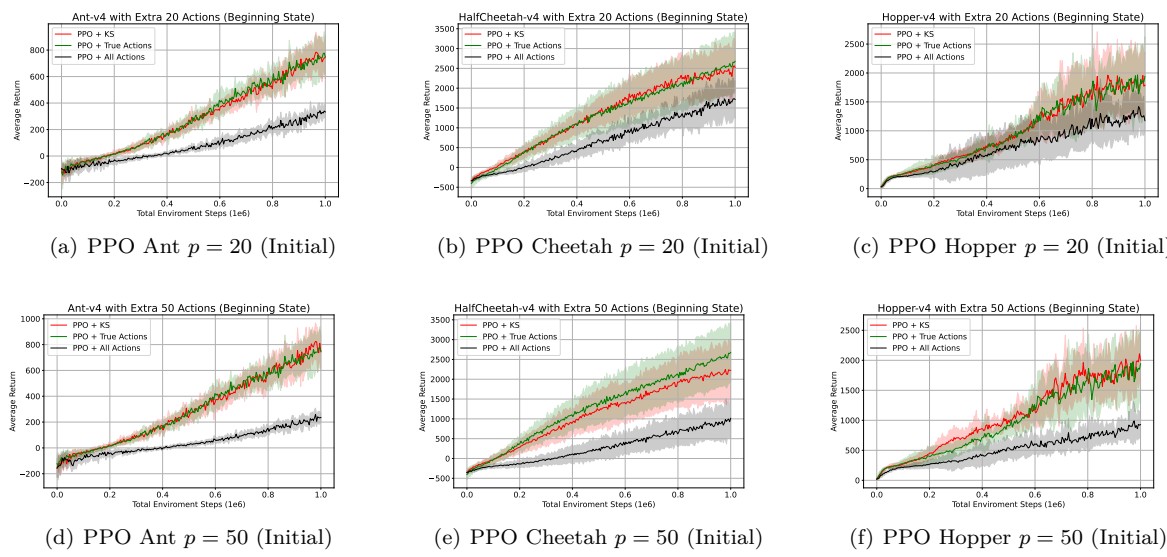

(a) PPO Ant $p = 20$ (Initial)          (b) PPO Cheetah $p = 20$ (Initial)          (c) PPO Hopper $p = 20$ (Initial)

(d) PPO Ant $p = 50$ (Initial)          (e) PPO Cheetah $p = 50$ (Initial)          (f) PPO Hopper $p = 50$ (Initial)

Figure 3: Learning curves for PPO in the MujoCo environments with different approaches during the initial stage. In all experiments, our knockoff sampling (KS) method not only performs comparably to the true actions but also consistently delivers higher rewards than using all actions.

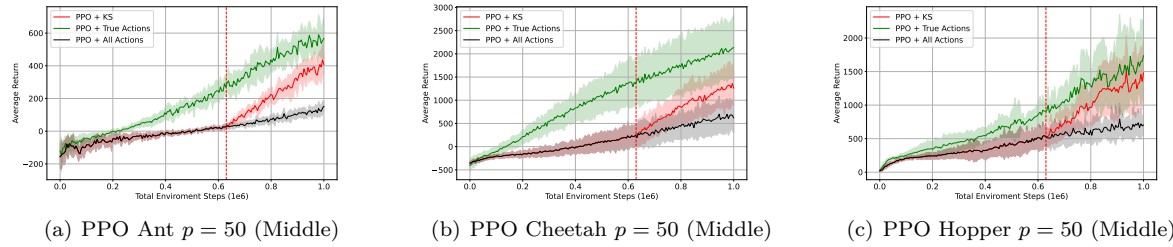

(a) PPO Ant $p = 50$ (Middle)          (b) PPO Cheetah $p = 50$ (Middle)          (c) PPO Hopper $p = 50$ (Middle)

Figure 4: Learning curves for PPO in the MujoCo environments during the middle stage, where the red line indicates the time point we utilize the proposed KS. After identifying the essential action set, the policy can be more efficient and achieve higher rewards than continuing training on all actions.

PPO, respectively, averaged over 10 training runs. The running steps for SAC and PPO are set adaptively to obtain better exploration for each method and save computation costs, as the main goal is to show how action selection can improve sample efficiency rather than compare these two methods. For each evaluation point, we run 10 test trajectories and average their reward as the average return. Besides RL algorithm performance, we also evaluate variable selection performance in terms of True Positive Rate (TPR), False Positive Rate (FPR), and FDR.

**Action Selection in the Initial Stage of Training** We utilize action selection in the beginning stage of the training. For both methods, we utilize the first 4000 samples for variable selection and then use the selection results to build a hard mask for action in deep RL models. We compare our knockoff sampling (KS) method with the baseline of selecting all actions (All) to evaluate the impact of integrating a masking mechanism with a selection strategy in deep RL. We also provide the experimental results with only ground-truth actions selected (True) as a reference. To reduce the computational complexity, we choose LASSO (Tibshirani, 1996) as our base variable selection algorithm for KS. Here, selecting all actions (All) and ground-truth actions (True) are the cases where RL models are trained on the whole action space and minimal sufficient action space, respectively. Hence, the model corresponding to ground-truth action has smaller parameters than all other methods because its initialization is based on the minimal sufficient action set. The results are shown in Fig. 3 for PPO, Fig. C.1 for SAC, and Table 1 for all numerical details. Due to space constraints, we mainly present the PPO figures in the main text. In all cases, we find that KS-guided models outperform

Table 1: Results on the PPO and SAC for three Mujoco tasks: Ant, HalfCheetah, and Hopper. Action selection is utilized at the beginning stage of RL training. The final reward is the performance evaluation for the agent after training. The best-performing results between KS and All are highlighted in bold.

| Env | RL Algo. | $p$ | Selection | Ant | | | |
|---|---|---|---|---|---|---|---|
| | | | | TPR ($\uparrow$) | FDR ($\downarrow$) | FPR ($\downarrow$) | Reward ($\uparrow$) |
| Ant | PPO | 0 | True | 1.00 | 0.0 | 0.00 | 567.77 |
| | | 20 | KS | 1.00 | **0.01** | **0.01** | **507.90** |
| | | | All | 1.00 | 0.71 | 1.00 | 202.65 |
| | | 50 | KS | 1.00 | **0.00** | **0.00** | **572.39** |
| | | | All | 1.00 | 0.86 | 1.00 | 151.66 |
| | SAC | 0 | True | 1.00 | 0.00 | 0.00 | 817.95 |
| | | 20 | KS | 1.00 | **0.01** | **0.01** | **937.74** |
| | | | All | 1.00 | 0.71 | 1.00 | 12.61 |
| | | 50 | KS | 1.00 | **0.00** | **0.00** | **731.73** |
| | | | All | 1.00 | 0.86 | 1.00 | $-208.04$ |
| HalfCheetah | PPO | 0 | True | 1.00 | 0.0 | 0.00 | 2130.55 |
| | | 20 | KS | 1.00 | **0.01** | **0.01** | **2237.08** |
| | | | All | 1.00 | 0.77 | 1.00 | 1356.46 |
| | | 50 | KS | 1.00 | **0.00** | **0.00** | **1932.27** |
| | | | All | 1.00 | 0.89 | 1.00 | 619.67 |
| | SAC | 0 | True | 1.00 | 0.00 | 0.00 | 6640.05 |
| | | 20 | KS | 1.00 | **0.00** | **0.00** | **6607.55** |
| | | | All | 1.00 | 0.77 | 1.00 | 5631.20 |
| | | 50 | KS | 1.00 | **0.00** | **0.00** | **6873.95** |
| | | | All | 1.00 | 0.89 | 1.00 | 4748.24 |
| Hopper | PPO | 0 | True | 1.00 | 0.0 | 0.00 | 1736.65 |
| | | 20 | KS | 1.00 | **0.00** | **0.00** | **1540.83** |
| | | | All | 1.00 | 0.87 | 1.00 | 1205.12 |
| | | 50 | KS | 1.00 | **0.00** | **0.00** | **1710.82** |
| | | | All | 1.00 | 0.94 | 1.00 | 703.08 |
| | SAC | 0 | True | 1.00 | 0.00 | 0.00 | 2511.00 |
| | | 20 | KS | 1.00 | **0.00** | **0.00** | **2165.81** |
| | | | All | 1.00 | 0.87 | 1.00 | 398.75 |
| | | 50 | KS | 1.00 | **0.00** | **0.00** | **2424.09** |
| | | | All | 1.00 | 0.94 | 1.00 | 137.67 |

those trained on the whole action space in terms of average return and have much lower FDR and FPR, with larger improvement gains as $p$ increases. This empirically validates our theory of FDR control with the proposed KS method, demonstrating that action selection can enhance learning efficiency during the initial stages of RL training where action space is high and redundant.

**Action Selection in the Middle Stage of Training** To show whether action selection can be used in the middle stage of training to remedy the inefficiency brought by exploring the whole action space, we conduct experiments where in the first half of the training steps the models are trained on the whole action space, and in the middle of the stage, we utilize action selection and build hard masks for them and then continue training for the rest of the steps. We compare our KS method with selecting all actions (All) and ground-truth actions (True) similarly. The results in Fig. 4 and Fig. C.2 reveal a notable pattern: agents initially struggle to learn effectively, but mid-stage variable selection significantly improves their performance, with models trained on the correct actions. This demonstrates the effectiveness of mid-stage variable selection in enhancing learning outcomes.

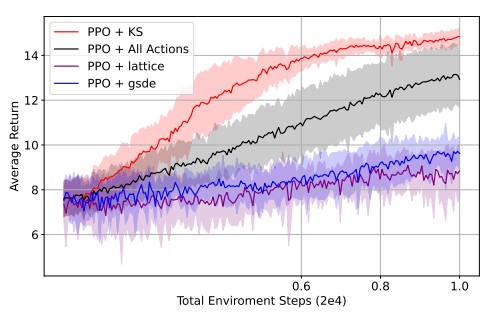 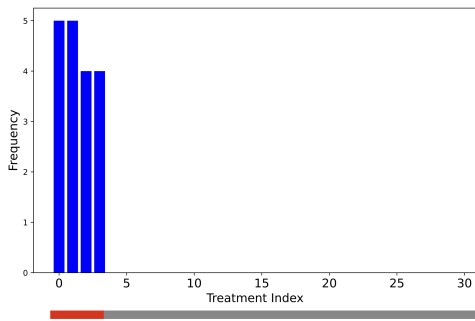

(a) Learning curves of treatment allocation environments.   (b) Treatment Selection frequency of KS method

Figure 5: Results for PPO in the treatment allocation environments during the initial stage using different approaches. The learning curves of various methods are shown on the left, while the selection frequencies from our method are presented on the right, with red bars indicating the essential treatments. Our approach, KS, demonstrates improved performance over time compared to using all actions. In contrast, latent exploration-based methods such as Lattice and gSDE exhibit degraded performance.

**Treatment Allocation for Sepsis Patients**  We evaluate our method using PPO and utilize the first 1000 samples for action selection during the initial stage of training. In addition to our proposed KS method and the baseline that uses all actions, we include two latent exploration-based baselines: Lattice (Chiappa et al., 2024) and gSDE (Raffin et al., 2022). While both Lattice and gSDE operate over the full action space, they incorporate temporally correlated Gaussian noise into the training process, where the noise variance is learned from latent representations. Experiments are conducted over $2 \times 10^4$ time steps, with results averaged across 5 independent runs. At each evaluation point, we generate 5 test trajectories and report the average return. The results are presented in Fig. 5(a) and Fig. 5(b). Our method consistently achieves more stable and superior performance compared to all other approaches in Fig. 5(a). As shown in Fig. 5(b), KS effectively identifies treatments directly relevant to sepsis management, such as `vaso_dose` and `iv_input`, which are closely tied to key physiological indicators. Importantly, KS avoids selecting those non-essential treatments like `beta_blocker` and `diuretic`, which may influence patient dynamics but have no direct effect on the SOFA score, making them less relevant for optimizing sepsis-specific outcomes. In contrast, Lattice and gSDE exhibit slower convergence and, in some cases, degraded performance, potentially due to over-exploration. We also observe that these methods are sensitive to the initialization of the log standard deviation and the scaling of latent representations, which can lead to unstable learning dynamics. In comparison, our method requires less parameter tuning and demonstrates greater robustness across different environments. These findings highlight the potential of KS for enabling more targeted, interpretable, and efficient treatment strategies in real-world medical decision-making applications.

**Action Selection is Fast and Lightweight**  With just a few thousand data points and a lightweight machine learning algorithm like random forest or LASSO, the whole action selection process outlined in Algorithm 2, completes in under 20 seconds—including knockoff threshold determination. This is significantly faster and less computationally intensive than the RL training part. Even when incorporating more sophisticated feature selection methods, the additional computational overhead remains *negligible* compared to the time required for RL training. Moreover, for the RL agent's deep neural network, only a few lightweight masking parameters are introduced, which have minimal effects on both training and inference speed. Yet, these in turn substantially enhance policy optimization.

**Additional Experiments**  We conduct additional experiments to visualize the action distribution during training, both with and without masking. The results indicate that masking promotes a more focused and potentially more effective learning process. Furthermore, we increase the PPO step size from $10^6$ to $4 \times 10^6$, demonstrating that our method achieves both high efficiency and improved performance. Additionally, we investigate whether network capacity plays a critical role in addressing high-dimensional action problems. However, we find that merely increasing network capacity does not necessarily simplify the learning process. Detailed results can be found in Appendix D.

## 6  Conclusion, Limitation, and Future Work

In this work, we address the high-dimensional action selection problem in online RL. We formally define the objective of action selection by identifying a minimal sufficient action set. We innovate by integrating a knockoff-sampling variable selection into broadly applicable deep RL algorithms. Empirical evaluations in synthetic robotics and treatment allocation environments demonstrate the enhanced efficacy of our approach. Yet, a notable constraint of our method is its singular application during the training phase, coupled with the potential risk of overlooking essential actions with weak signals. Inadequate action selection could degrade the agent's performance. Intriguing future research includes extending our methodology to incorporate multiple and adaptive selection stages. This adaptation could counterbalance initial omissions in action selection. Additionally, formulating an effective termination criterion for this process represents another compelling research direction.

## Acknowledgements

The authors thank the Action Editor and anonymous reviewers for their constructive and insightful feedback. This work was supported by the National Science Foundation under grant DMS-CDS&E-MSS No. 2401271.

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

# A    Comparison with Closely Related Work

Ma et al. (2023) adopted a two-stage framework, performing variable selection offline before applying reinforcement learning. This design is ill-suited for online settings, where dynamic environments require frequent updates to the action set. Moreover, their approach relies on strong data-generating assumptions that restrict its applicability in realistic scenarios. In contrast, our method integrates action selection into the online RL process via a learned masking mechanism, allowing real-time adaptation to environmental shifts. This not only removes the dependency on stringent assumptions but also supports seamless integration with deep RL algorithms, such as PPO and SAC, which were not addressed in their work.

Zahavy et al. (2018) introduced an action elimination method that relies on explicit elimination signals from the environment to identify invalid actions. While effective in certain settings, this approach depends on domain-specific feedback that may not always be available. Our method avoids this reliance by implicitly identifying essential actions through learned representations, enabling broader applicability without handcrafted elimination signals or external supervision.

Zhong et al. (2024) proposed a method that constructs an action mask using an inverse dynamics model trained prior to policy learning. However, fitting such a model is computationally demanding in high-dimensional state spaces and is completely decoupled from the learning process, making it less responsive to evolving environments. Additionally, extending their approach—or Zahavy et al. (2018)—to continuous action spaces typically requires discretization, which introduces challenges such as loss of precision and the need to carefully tune bin sizes. In contrast, our method operates directly in the continuous action space, preserving precision and scalability while enabling efficient, end-to-end learning.

# B    Implementation Details

We adopt the implementation from Open AI Spinning up Framework (Achiam, 2018). Tables B.1 and B.2 show the hyperparameters for the RL algorithms we used in our experiments. We set the FDR rate $\alpha = 0.1$ and voting ratio $r = 0.5$ for our knockoff method in all settings. All the experiments are conducted in the server with $4\times$ NVIDIA RTX A6000 GPU.

Table B.1: PPO Hyperparameters

| Parameter | Mujoco | EHR |
|---|---|---|
| optimizer | Adam | Adam |
| learning rate $\pi$ | $3.0 \cdot 10^{-4}$ | $3.0 \cdot 10^{-3}$ |
| learning rate V | $1.0 \cdot 10^{-3}$ | $1.0 \cdot 10^{-3}$ |
| learning rate schedule | constant | constant |
| discount ($\gamma$) | 0.99 | 0.99 |
| number of hidden layers (all networks) | 2 | 2 |
| number of hidden units per layer | $[64, 32]$ | $[64, 32]$ |
| number of samples per minibatch | 256 | 100 |
| number of steps per rollout | 1000 | 100 |
| non-linearity | ReLU | ReLU |
| gSDE | | |
| initial log $\sigma$ | 0 | 0 |
| Full std matrix | Yes | Yes |
| Lattice | | |
| initial log $\sigma$ | 0 | 0 |
| Full std matrix | Yes | Yes |
| Std clip | (0.001,1) | (0.001,1) |

Table B.2: SAC Hyperparameters

| Parameter | Mujoco |
|---|---|
| optimizer | Adam |
| learning rate $\pi$ | $3.0 \cdot 10^{-4}$ |
| learning rate Q | $3.0 \cdot 10^{-4}$ |
| learning rate schedule | constant |
| discount ($\gamma$) | 0.9 |
| replay buffer size | $1 \cdot 10^6$ |
| number of hidden layers (all networks) | 2 |
| number of hidden units per layer | $[256, 256]$ |
| number of samples per minibatch | 256 |
| non-linearity | ReLU |
| entropy coefficient ($\alpha$) | 0.2 |
| warm-up steps | $1.0 \cdot 10^4$ |

Table C.1: Summary of Environments

| Env | Dimension of Action | Dimension of State |
|---|---|---|
| Ant | 8 | 27 |
| HalfCheetah | 6 | 17 |
| Hopper | 3 | 11 |
| EHR | 20 | 46 |

## C   More Experimental Results and Analyses

We list the dimension of action and state in terms of the environments we used in Table C.1.

### C.1   MuJoCo

The results for the initial stage are shown in Fig. C.1 and Table 1. For different environments, action selection difficulty varies, and Hopper is the easiest one where the proposed KS method can correctly select the minimal sufficient action set. Also, the patterns of the results for PPO and SAC are similar. One observation is that KS can select all the sufficient actions with all TPRs equal to 1. It is shown that KS is efficient in selecting only the minimal sufficient action set in almost all scenarios, which also empirically validates our theory of FDR control under the proposed method.

### C.2   Treatment Allocation for Sepsis Patients

We utilize the MIMIC-III Clinical Database to construct our environment for Sepsis patients. We filter and clean the data to collect $250,000$ data points from the database.

**State Space**. The observed state $\mathbf{s}_t \in \mathbb{R}^{46}$ encompasses a broad spectrum of clinical and laboratory variables for assessing patient health and outcomes in a medical setting, such as demographic information (gender, age), physiological metrics (weight, vital signs such as heart rate, blood pressure, respiratory rate, oxygen saturation, temperature), and neurological status. The transition is modeled by $\mathbf{s}_t = f_\theta(\mathbf{s}_{t-1}, \mathbf{a}_{t-1}) + \epsilon$, where $f_\theta$ is a fitted long short-term memory (LSTM) and $\epsilon$ is the random noise to represent the perturbation.

**Action Space**. The action $\mathbf{a}_t \in \mathbb{R}^{30}$ includes treatments such as vasopressors and intravenous fluids, which are commonly administered in sepsis management to stabilize blood pressure and maintain fluid balance, in total 4 actions. Additionally, we incorporate other medications, beta-blockers, and diuretics, which—although not primarily intended for sepsis—may be prescribed to manage comorbid conditions like hypertension or fluid overload that often coexist with or complicate sepsis. These treatments are considered less essential

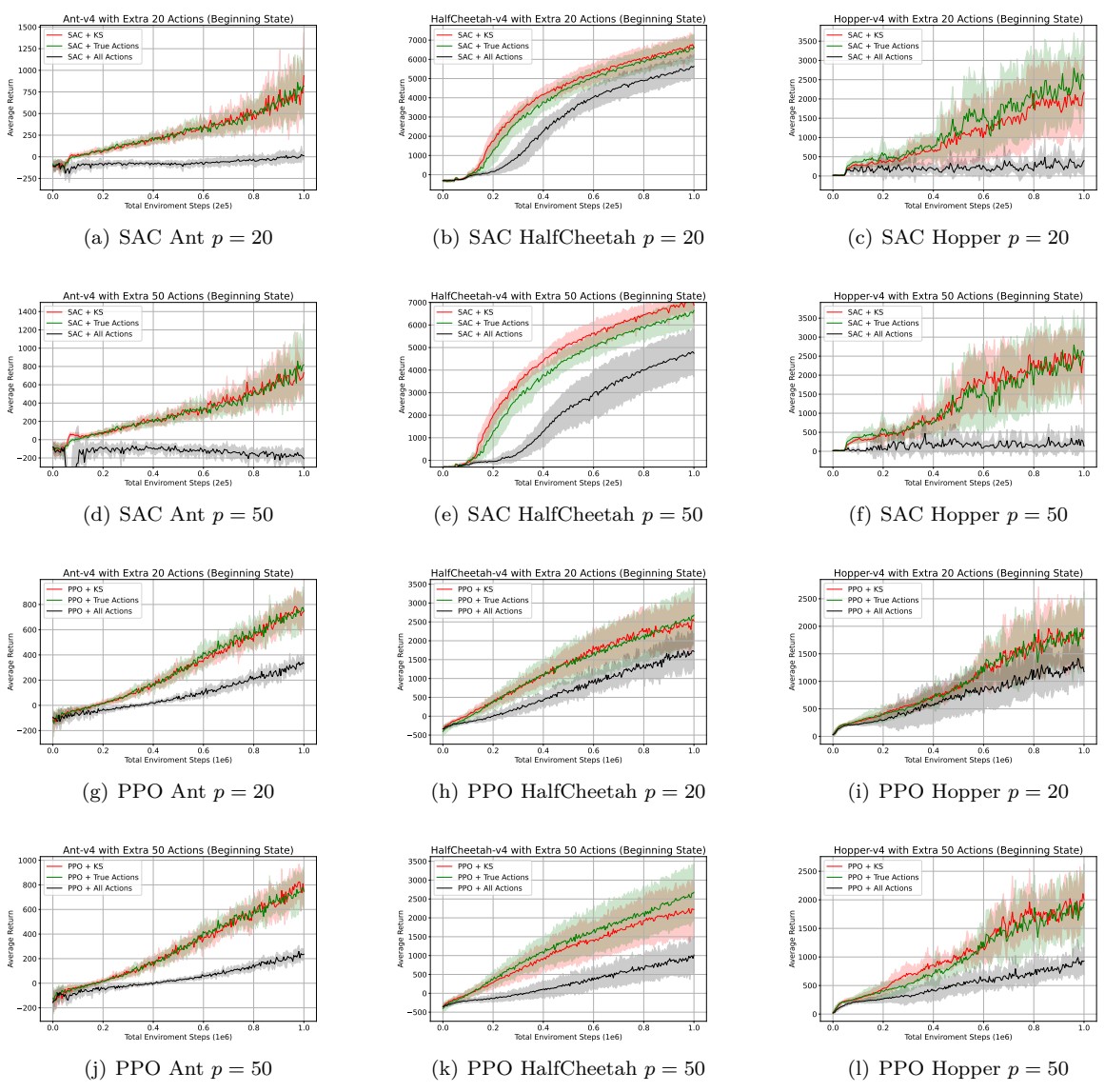

Figure C.1: Results of SAC and PPO when using different variable selection approaches during the initial stage.

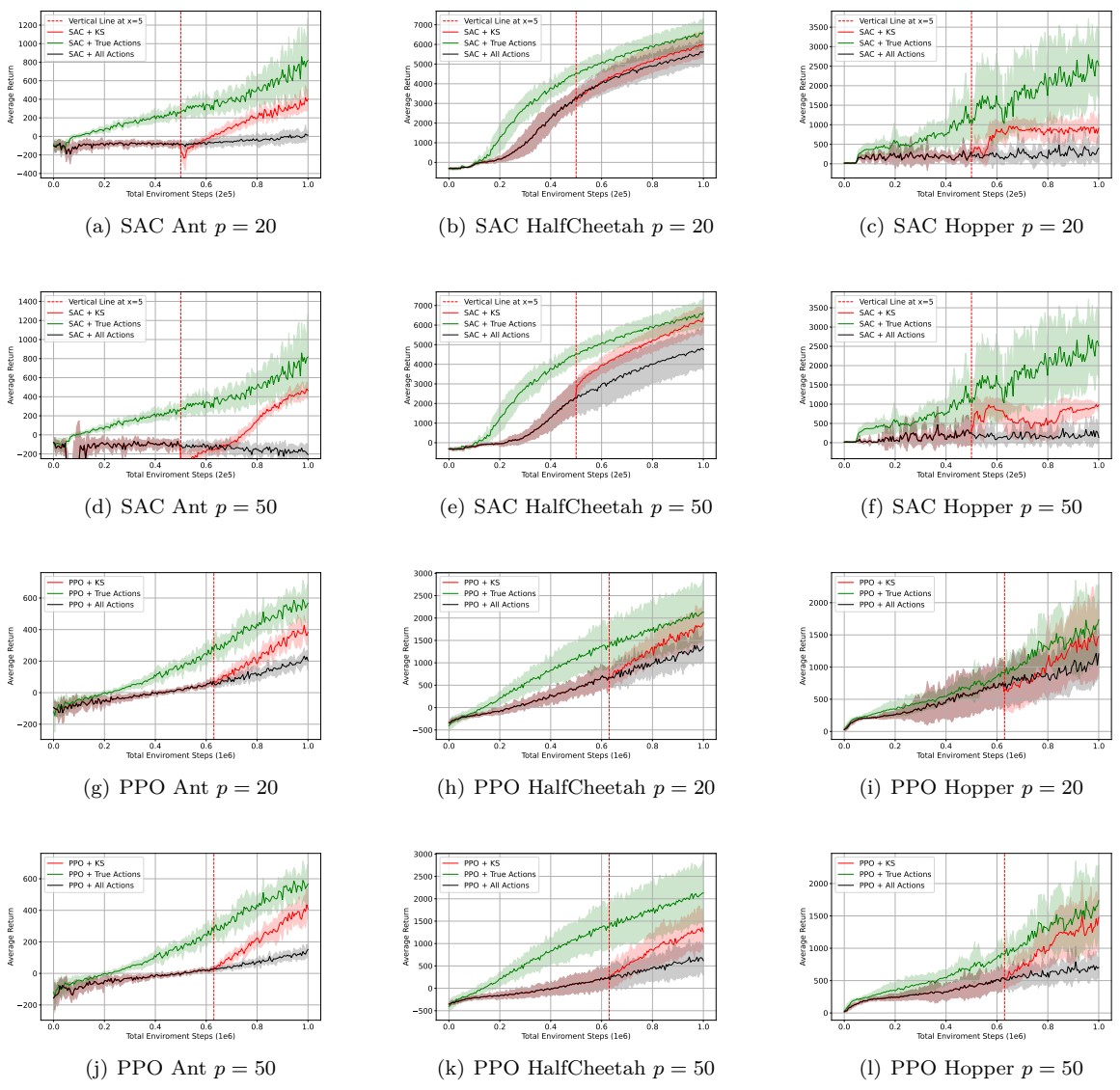

Figure C.2: Learning curves in the MujoCo environments with our method during the middle stage, where the red line indicates the time we utilize KS. After identifying a less redundant action set, the policy can be more efficient and achieve higher rewards than continuing training on all actions.

or potentially redundant in the context of sepsis management. To further increase the complexity of the environment, we augment the action space with treatments and medications that have minimal relevance to sepsis, thereby making the decision-making task more challenging.

**Reward Function**. The overall reward at each timestep is composed of two components, represented as $r_t = r_t^s + r_t^h$, where $r_t^s$ denotes the SOFA-based sepsis reward and $r_t^h$ captures a health-conditioned bonus. The sepsis reward $r_t^s$ reflects changes in the patient's SOFA score, assigning $+1.0$ if the score decreases (indicating clinical improvement), and $-1.0$ if the score increases (indicating deterioration). In addition to this primary reward signal, we introduce a health-conditioned bonus $r_t^h$ which provides supplementary guidance based on the patient's overall physiological state $s_t$ and the administered treatment $a_t$. This component allows the reward function to capture nuanced aspects of patient health beyond SOFA score dynamics.

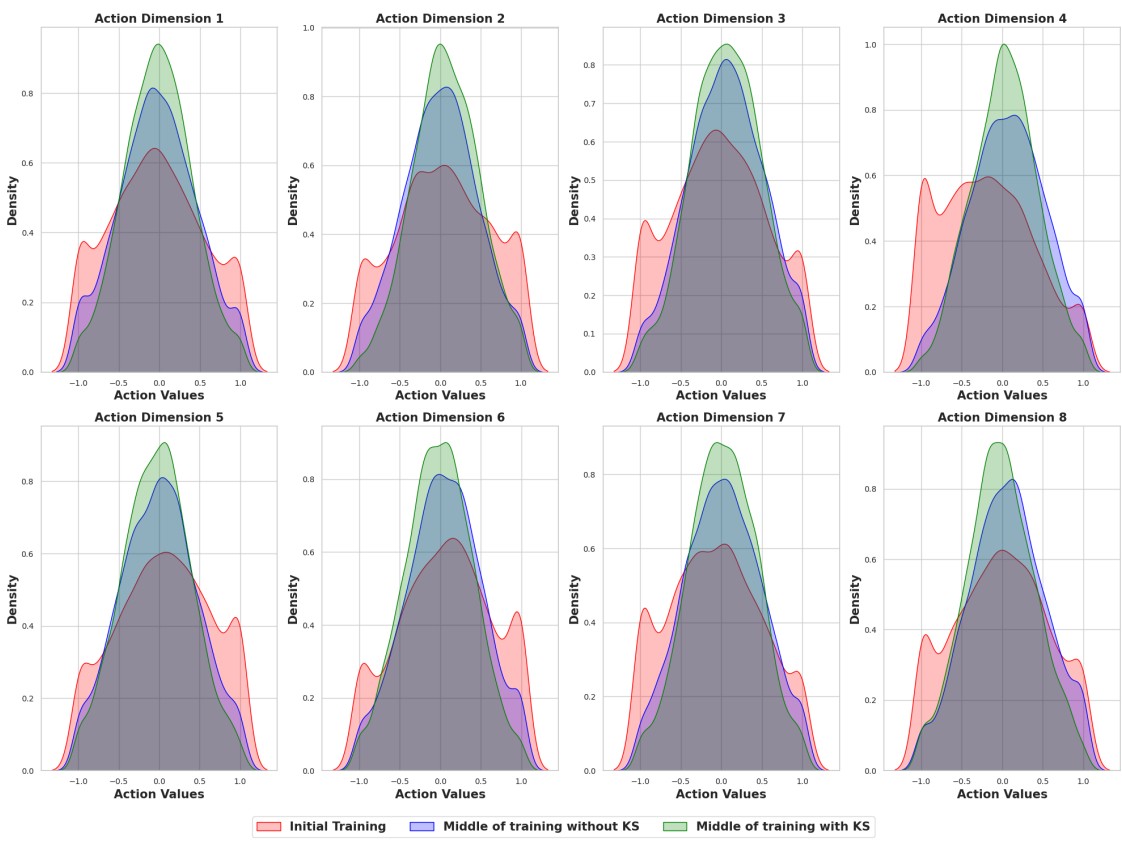

Figure D.1: The distributions of actions in 3 stages: initial training, middle of training without KS, and middle of training with KS. It can be seen that with KS, the actions have slightly less variance than other methods, which could be a positive indicator of a more focused and potentially more effective learning process.

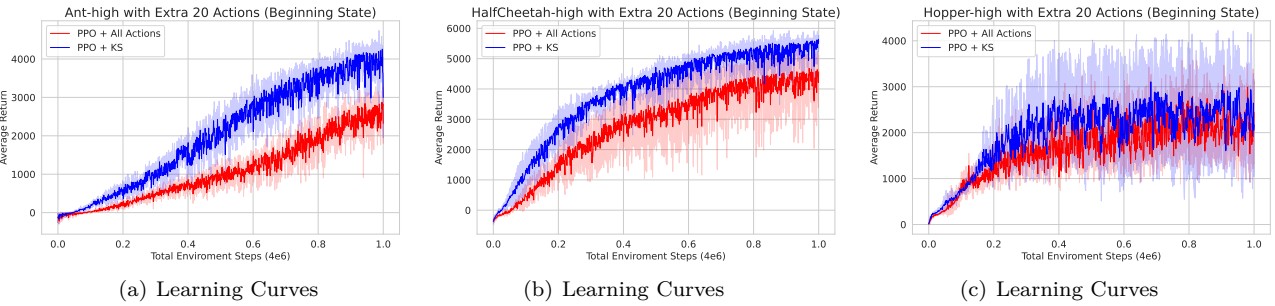

Figure D.2: Learning curves in the MujoCo environments with $4e6$ steps.

Termination of an episode is achieved based on the patient's mortality rate reaching the minimum (SOFA Score being 0) or the patient's mortality rate reaching the maximum (SOFA score being 24). We also observe that Weight-kg and cumulated-balance have minimal influence on Sepsis; therefore, we treat them as non-essential state variables and exclude them when constructing the response $\mathbf{y}_t$ for variable selection.

Table C.2: List of state and action variables with their corresponding indices in the Treatment Allocation Environment for Sepsis Patients. Here, red indicates essential treatments, and light red represents non-essential treatments and ineffective treatments.

| State Index | State Name | Action Index | Action Name |
|---|---|---|---|
| 0 | gender | 0 | vaso_dose_1 |
| 1 | age | 1 | vaso_dose_2 |
| 2 | elixhauser | 2 | vaso_dose_3 |
| 3 | re_admission | 3 | iv_input |
| 4 | Weight_kg | 4 | beta_blocker |
| 5 | GCS | 5 | diuretic |
| 6 | HR | 6 | antihistamine |
| 7 | SysBP | 7 | proton_pump_inhibitor |
| 8 | MeanBP | 8 | statin |
| 9 | DiaBP | 9 | metformin |
| 10 | RR | 10 | calcium_channel_blocker |
| 11 | SpO2 | 11 | antidepressant |
| 12 | Temp_C | 12 | antipsychotic |
| 13 | FiO2_1 | 13 | antacid |
| 14 | Potassium | 14 | levothyroxine |
| 15 | Sodium | 15 | NSAID |
| 16 | Chloride | 16 | laxative |
| 17 | Glucose | 17 | multivitamin |
| 18 | BUN | 18 | topical_ointment |
| 19 | Creatinine | 19 | cough_suppressant |
| 20 | Magnesium | 20 | homeopathic_remedy |
| 21 | Calcium | 21 | herbal_supplement |
| 22 | Ionised_Ca | 22 | eye_drops |
| 23 | CO2_mEqL | 23 | muscle_relaxant |
| 24 | SGOT | 24 | antihypertensive_class_B |
| 25 | SGPT | 25 | sleep_aid |
| 26 | Total_bili | 26 | nasal_decongestant |
| 27 | Albumin | 27 | acne_treatment |
| 28 | Hb | 28 | antiemetic |
| 29 | WBC_count | 29 | vitamin_D_supplement |
| 30 | Platelets_count | | |
| 31 | PTT | | |
| 32 | PT | | |
| 33 | INR | | |
| 34 | Arterial_pH | | |
| 35 | paO2 | | |
| 36 | paCO2 | | |
| 37 | Arterial_BE | | |
| 38 | Arterial_lactate | | |
| 39 | HCO3 | | |
| 40 | mechvent | | |
| 41 | Shock_Index | | |
| 42 | PaO2_FiO2 | | |
| 43 | cumulated_balance | | |
| 44 | SOFA | | |
| 45 | SIRS | | |

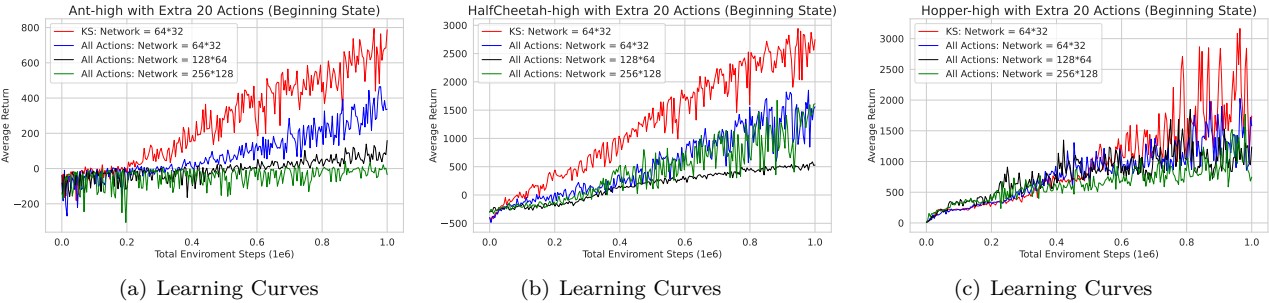

(a) Learning Curves  (b) Learning Curves  (c) Learning Curves

Figure D.3: Learning curves in the MujoCo environments with different network sizes.

## D  Supporting Analyses

We conducted additional experiments regarding the distribution of actions that were sampled over the training period. The new results are summarized in Figure D.1. Based on the results together with existing figures in the main text, we can conclude that there exist changes regarding the distribution of actions that are sampled during different periods of training, which could be a positive indicator of a more focused and potentially more effective learning process.

Since we only use steps for PPO training, which might be able to converge in the end, we further add the steps to 4e6. The new results are summarized in Figure D.2. Based on the results together with existing figures in the main text, we can conclude that the benefit of the proposed framework is both sample efficiency and performance.

We also conducted additional experiments by increasing the network size. The new results under MujoCo with different network sizes are summarized in Figure D.3, where we can conclude that the network capacity has a certain influence on the performance of All Action. However, simply increasing the network capacity unnecessarily makes learning easier. In addition, the proposed method consistently performs better than All Action, no matter how we increase the network size, which indicates that the performance difference results from the redundancy of the action space. In addition, we admit there are other factors that affect how the agent learns with a larger action dimension, such as regularization techniques, albeit with limited influence compared with the redundancy of the action space.

## E  Dimensionality Reduction and Estimation Bias

A common concern in feature or action space reduction is the potential introduction of bias due to the restriction of the hypothesis space, particularly when informative variables are inadvertently excluded. In our setting, however, the redundant actions are conditionally independent of both the reward and the next state, and thus can be regarded as nuisance variables that do not contribute meaningful information to the decision-making process. When these actions are correctly identified and excluded, the resulting reduction in dimensionality does not increase bias in principle. In fact, including irrelevant or weakly related features may introduce additional bias and variance by complicating the model-fitting process. Moreover, from a theoretical perspective, reducing the input dimensionality is beneficial for both bias and variance. According to established results in deep learning theory (Farrell et al., 2021), the error bound for value or policy estimation using multilayer perceptrons is on the order of $O(n^{-\beta/(\beta+d)})$, where $d$ denotes input dimensionality, $n$ is the sample size, and $\beta$ is the Hölder smoothness parameter. Thus, selecting a minimal set of sufficient actions and removing redundant ones leads to improved sample efficiency and more accurate value function approximation.

## F    Extend to Correlated Actions

Now, assume the policy network is parameterized by a multivariate Gaussian with a covariance matrix as:

$$\mathbf{a} \sim \mathcal{N}\left(\mu\left(\mathbf{s}\right), \Sigma\left(\mathbf{s}\right)\right),$$

where $\mu$ and $\Sigma$ are parameterized functions to output the mean and covariance matrix.

Since the actions are correlated, we couldn't mask the individual $\log \pi_\theta\left(a_i \mid \mathbf{s}\right)$. To solve this problem, we can transform the non-selected actions to be conditional independent first.

Given a selected action set $\widehat{G}$, we define a selection vector $\mathbf{m} = (m_1, \cdots, m_p) \in \{0, 1\}^p$, where $m_i = 1$ if $i \in \widehat{G}$ and 0 otherwise. Then we mask the covariance matrix as follows:

$$\Sigma^m(\mathbf{s})_{ij} \begin{cases} \Sigma(\mathbf{s})_{ij} & \text{if } m_i = 1, \text{and } m_j = 1, i \neq j \\ 0 & \text{if } m_i = 0, \text{or } m_j = 0, i \neq j. \end{cases}$$

The masking only changes non-selected actions to be independent and removes their influence on selected actions, which still keeps the covariance structure of selected actions. Then we can easily mask the log density of non-selected actions.

## G    Machine Learning Algorithm for Calculating Importance Scores

The feature importance score $Z_{i,j}$ for the $i-$th outcome (the reward or the next state) and the $j-$th action is computed by fitting this outcome based on all inputs using a machine learning method. Specifically, this process fits a predictive model $\hat{f}(x)$ to estimate a target variable $y$, where $y$ corresponds to either the reward or the next state, the feature vector $x$ comprises the current state, action, and a knockoff copy, and the function $f$ can be the function classes of LASSO, random forests, or neural networks. The importance score of a specific variable corresponds to its estimated coefficient in LASSO, its feature importance score in random forests, or its weight/gradient in neural networks. Separate models are constructed for each target outcome as detailed in Algorithm 2.

The only requirement for the machine learning method is that it satisfies a fairness constraint, ensuring that exchanging an original feature with its knockoff counterpart results solely in the corresponding exchange of the model's importance score for those features. This condition is typically met by standard tabular machine learning algorithms.

## H    Extension to Non-stationary Environments

In the context of our paper, stationarity refers to the policy being fixed and the transition dynamics and reward function remaining unchanged. Hence, in the common batch update setting, the data collected between policy updates can be treated as stationary. Our method can accommodate different forms of non-stationarity.

When non-stationarity arises solely due to policy updates—while the environment's dynamics and rewards remain unchanged—our method can effectively handle this scenario by conducting knockoff variable selection within each batch. This approach boosts policy optimization by ensuring that action masking is informed by relevant, stable data. To enhance robustness, we can also extend our selection procedure to operate every k batch, leading to more stable and adaptive masking throughout training.

In cases where non-stationarity stems from changes in the transition dynamics or reward function—which is rare in simulated environments like MuJoCo but plausible in real-world scenarios—the problem becomes more challenging. Nonetheless, our method remains effective under the realistic assumption of piecewise stationarity. By integrating change point detection, we can segment the data into stationary intervals and apply feature selection within each segment. This ensures that the selected action subset remains relevant and informative, thereby preserving and even enhancing policy optimization.

# I Technical Proofs

## I.1 Preliminary Results

Before we prove Theorem 1, we first provide a preliminary lemma of our procedure that can enable the flip-sign property of W-statistics. This property can be used to prove Theorem 1 when observations are independent. Now we focus on one data split $\mathcal{D}_k$ and assume the data are independent.

**Lemma I.1.** $\mathbf{A}_k$ and $\mathbf{S}_k$ are a action and a state matrix. For any subset $\Omega \subset \{1, \ldots, p\}$, and $\tilde{\mathbf{A}}_k$ obatined by resampling, we have

$$\left( \left[ \mathbf{A}_k, \tilde{\mathbf{A}}_k \right]_{\mathrm{swap}(\Omega)}, \mathbf{S}_k \right) \overset{d}{=} \left( \left[ \mathbf{A}_k, \tilde{\mathbf{A}}_k \right], \mathbf{S}_k \right),$$

where $\mathrm{swap}(\Omega)$ represents swapping the j-th entry of $\mathbf{A}_k$ and $\tilde{\mathbf{A}}_k$ for all $j \in \Omega$.

The proof of Lemma I.1 is based on the property of constructed variables where $\mathbf{A}_k$ and $\tilde{\mathbf{A}}$ have the same marginal distribution and the whole joint distribution is symmetrical in terms of $\mathbf{A}_k$ and $\tilde{\mathbf{A}}$.

In the following, we show that the exchangeability holds jointly on actions, states, and rewards when swapping null variables.

**Lemma I.2.** Let $\mathcal{H}_0 \subseteq \{1, \ldots, p\}$ be the indices of the null variables, for any subset $\Omega \subset \mathcal{H}_0$

$$\left( \left[ \mathbf{A}_k, \tilde{\mathbf{A}}_k \right]_{\mathrm{swap}(\Omega)}, \mathbf{S}_k, \mathbf{Y}_k \right) \overset{d}{=} \left( \left[ \mathbf{A}_k, \tilde{\mathbf{A}}_k \right], \mathbf{S}_k, \mathbf{Y}_k \right),$$

where $\mathbf{Y}_k$ is a response including the next state and reward.

**Proof:** Based on the exchangeability proved in Lemma I.1, we can directly utilize the proof of Lemma 3.2 in Candès et al. (2018). We just need to extend the derivation by conditioning on $\mathbf{S}_k$ and show equivalence by swapping action variables in $\Omega$ one by one. We omit further details of the proof.

**Lemma I.3.**

$$W_i \left( \left[ \mathbf{A}_k, \tilde{\mathbf{A}}_k \right]_{\mathrm{swap}(\Omega)}, \mathbf{S}_k, \mathbf{Y}_k \right) = W_i \left( \left[ \mathbf{A}_k, \tilde{\mathbf{A}}_k \right], \mathbf{S}_k, \mathbf{Y}_k \right) \cdot \begin{cases} -1, & \textit{if } i \in \Omega \\ +1, & \textit{otherwise} \end{cases}.$$

**Proof:** We require the method for constructing $W$ to satisfy a fairness requirement so that swapping two variables would have the only effect of swapping corresponding feature importance scores. The fairness constraint is satisfied with many general machine learning algorithms, like LASSO and random forest. Once the fairness constraint is satisfied, $W$ will be anti-symmetric, and the equality above automatically holds.

**Lemma I.4.** *Assume the flip-coin property in Lemma I.3 is satisfied, on data $\mathcal{D}_k$, the selection $\widehat{G}_k$ obtained from applying knockoff method in Algorithm 2 controls modified FDR (mFDR), e.g.*

$$mFDR \left( \widehat{G}_k \right) \leq \alpha.$$

**Proof:** For statistics $W$ calculated, we denote $W_{\mathrm{swap}(\Omega)}$ to be the $W$-statistics computed after the swap w.r.t. $\Omega \subset \{1, \ldots, p\}$. Now consider a sign vector $\epsilon \in \{\pm 1\}^p$ independent of $\mathbf{W} = [W_1, \ldots, W_p]^\top$, where $\epsilon_i = 1$ for all non-null state variables and $\mathbb{P}(\epsilon_i = 1) = 1/2$ are independent for all null state variables. Then for such $\epsilon$, denote $\Omega := \{i : \epsilon_i = -1\}$, which is a subset of $\mathcal{H}_0$ by the assumption (and recall that $\mathcal{H}_0$ is the collection of all null variables). By Lemma I.3 we know

$$(W_1 \cdot \epsilon_1, \ldots, W_p \cdot \epsilon_p) = W_{\mathrm{swap}(\Omega)}.$$

For convenience, we also use $h$ to denote a measurable mapping function from a data set to its $W$-statistics, i.e., on $[\mathbf{s}_k, \tilde{\mathbf{s}}_k, \mathbf{a}_k, \mathbf{y}_k]$,

$$\mathbf{W} = h \left( \mathbf{A}_k, \tilde{\mathbf{A}}_k, \mathbf{S}_k, \mathbf{Y}_k \right).$$

Then we can get:

$$\mathbf{W}_{\mathrm{swap}(\Omega)} = h\left(\left[\mathbf{A}_k, \tilde{\mathbf{A}}_k\right]_{\mathrm{swap}(\Omega)}, \mathbf{S}_k, \mathbf{Y}_k\right)$$
$$\overset{d}{=} h\left(\left[\mathbf{A}_k, \tilde{\mathbf{A}}_k\right], \mathbf{S}_k, \mathbf{Y}_k\right) = \mathbf{W},$$

where the second equality (in distribution) is due to Lemma I.2 and $h$ is measurable. The rest of the proof will be the same as that for Theorems 1 and 2 in Barber & Candès (2015).

### I.2    Proof of Theorem 4.4

Using Lemma I.4, we can show that if the data points in $\mathcal{D}_k$ are independent, then $mFDR$ can be controlled. Now we want to weaken the independence assumption to stationarity and exponential $\beta$-mixing assumption in 4.3. Based on Lemma I.4, the following proof is essentially the same as Theorem 1 in Ma et al. (2023). We will omit those steps for brevity.

