# OpenReview forum: "Where to Intervene: Action Selection in Deep Reinforcement Learning"
_TMLR — Accepted by TMLR_

### Review · Reviewer_3oS8 · 2024-12-08

**Summary Of Contributions:**

This paper addresses the issues resulting from large action spaces with
redundant components in reinforcement learning. The authors argue
naively applying RL methods in these settings leads to needless increase
in sample complexity, and they propose methods, using tools such as
*knockoff sampling* from the variable selection literature, for
automatically discovering these redundant action components so they can
be pruned. Results show that their method successfully identifies
redundant action components, leading to sample complexity rivalling
equivalent systems with a minimum viable action space.

**Audience:**

Yes

**Broader Impact Concerns:**

No broader impact concerns.

**Claims And Evidence:**

Yes

**Requested Changes:**

1.  There is no discussion about what "false discovery rate" (FDR)
    means. I don't know what this means, so I am having a hard time
    following the paper.
2.  Section 2.3 requires much more detail. For example,
    1.  What is a "response vector"? Are these essentially labels for a
        supervised learning problem? Perhaps this would simply be
        cleared up again with a description of FDR / the setting you
        consider in this section.
    2.  Equation (1) says $(\pmb{X},
                \widetilde{\pmb{X}})_{\mathrm{swap}(\Omega)}\overset{d}{=}(\pmb{X},
                \widetilde{\pmb{X}})$, but in your definitions of these
        variables/operations, it is entirely unclear to me where there
        is any randomness.
    3.  How do you calculate feature importance scores?
    4.  It says "set $W_j = f(Z_j, \widetilde{Z}_j)$ in such a way that
        higher values of $W_j$ indicate stronger evidence of $\pmb{x}_j$
        being influential covariate". Isn't this entirely dictated by
        the choice of $f$? How should one choose $f$? How do you know
        chich covariates are likely to be influential so as to design
        $f$ to satisfy this condition?
    5.  Again, it would be very helpful if the final result (e.g.
        equation (2)) could be contextualized. What is accomplished with
        this filtering rule?
3.  Under equation (3), it says "action selection reduces the
    dimensionality of the input action space, there by reducing bias in
    $Q$ function fitting". This warrants some more discussion,
    intuitively I would expect the opposite, since you are effectively
    restricting the hypothesis class over $Q$ functions to a smaller one
    (more bias).
4.  In equation (4), essentially you're suggesting to stop policy
    gradients through action dimensions that are redundant. Why is this
    the correct strategy? How does this allow the policy to ignore those
    dimensions (e.g., perform "action selection")? Why don't you
    additionally act according to $\mathbf{m}\odot\mathbf{a}$ instead of
    $\mathbf{a}$ sampled under the policy (or alternatively, is there any way to concretely use the learned mask to enforce that the policy acting in a manner consistent with the unmasked actions only)?
5.  In definition 4.1, there appears to be a missing word or something
    at the beginning of the definition. There appears to be a sentence
    starting with "property on the augmented data matrix \[…\]".
6.  In Theorem 4.3, what is "modified FDR"? What is $\alpha$?
7.  There is a notable lack of discussion regarding Theorem 4.3.
    Probably exacerbated by the difficulty I had following the
    background section, it is very hard to intuit the consequences of
    this theorem. It should be clearly explained how this bound applies
    to RL / policies, and why we want systems under which this upper
    bound is minimized.
8.  In the "Semi-synthetic MuJoCo Environments", what does it mean to
    "add extra $p$ actions to the raw action space"? What are these
    action dimensions? Are they just ignored by the simulator?

**Strengths And Weaknesses:**

**Strengths**: The use of variable selection for identifying a minimal
set of actions is interesting, and in particular, the application of
knockoff sampling for this purpose appears to be novel and non-trivial.
The empirical results of the proposed method are reasonably good,
particularly I found the results of Figure 4 to be interesting.

**Weaknesses**: There are a few major weaknesses, in my opinion, which I
list below. Further details are given in the Requested Changes.

1.  The background on variable selection and knockoff sampling is
    extremely difficult to follow—I could not make any sense of it.
    Clarity of this section absolutely must be improved, since it plays
    a large role in the paper. One consequence, besides not having an
    intuitive understanding of the proposed algorithm, is that Theorem
    4.3 (the main theorem in the paper) is extemely difficult to
    comprehend and appreciate.
2.  The motivation of the problem was fairly weak, in my opinion. It was
    not clear to me until the algorithmic section that the goal was not
    to reduce the action space for the sake of, say, enabling a
    tractable class of policies. Indeed, from the perspective of the
    neural net architecture, the size of the action space doesn't
    change. This paper focuses entirely on the problem of identifying
    redundant actions to effectively improve sample efficiency. While I
    understand that this can lead to significantly accelerated training,
    I don't really know of any domains where there are so many redundant
    action features (the synthetic MuJoCo domains don't count). The
    authors highlight medical domains, but this is not discussed much in
    the experimental sections, and few concrete examples are given.
3.  Similarly to the previous point, the experiments could have been
    more convincing. While I liked Figure 4 as it shows basically how
    artificially adding redundant actions makes MuJoCo more difficult, I
    would have loved to see this figure for a domain that is more real.
    Are there every enough redundant actions that you can see a dramatic
    change like this? Where do these redundant actions arise from? Why
    couldn't we see a similar plot, for example, in the sepsis treatment
    domain?

---

> ### Author Response · Authors · 2025-04-07
>
> We are very grateful for your valuable comments and insightful feedback! In response to your comments, we make the following clarifications and additional revisions to the paper. Below, we summarize your comments in quotes and provide our point-by-point responses. Please refer to the latest submission for the revised paper, taking all your suggestions using **blue** color.
>
>
> ***Responses to Weaknesses***
>
> > 1. The background on variable selection and knockoff sampling is extremely difficult to follow—I could not make any sense of it. Clarity of this section absolutely must be improved, since it plays a large role in the paper. One consequence, besides not having an intuitive understanding of the proposed algorithm, is that Theorem 4.3 (the main theorem in the paper) is extemely difficult to comprehend and appreciate.
>
> Thank you for the valuable feedback. Following your suggestions, we have substantially revised Section 2.3 (Preliminary: Knockoff Variable Selection) to improve clarity regarding variable selection and the knockoff framework. Specifically, we’ve added more background on the principles of variable selection and included a detailed, step-by-step explanation of how knockoff statistics are constructed to make the methodology more accessible and easier to follow. Please also refer to our point-by-point responses addressing your specific questions in “Requested Changes” for more details.
>
>
> > 2. The motivation of the problem was fairly weak, in my opinion. It was not clear to me until the algorithmic section that the goal was not to reduce the action space for the sake of, say, enabling a tractable class of policies. Indeed, from the perspective of the neural net architecture, the size of the action space doesn't change. This paper focuses entirely on the problem of identifying redundant actions to effectively improve sample efficiency. While I understand that this can lead to significantly accelerated training, I don't really know of any domains where there are so many redundant action features (the synthetic MuJoCo domains don't count). The authors highlight medical domains, but this is not discussed much in the experimental sections, and few concrete examples are given.
>
> Thanks for your insightful comments. Per your suggestions, we’ve expanded the motivation in the introduction to emphasize that: “Our method does not reduce the dimensionality of the action space itself but rather focuses on identifying redundant actions and mitigating their influence during policy training to enhance sample efficiency and significantly accelerate learning with implementation and interpretation advantages”.
>
> In medical domains, when facing a new disease with limited prior knowledge, doctors often try multiple treatments to observe their effects, which can lead to redundant actions. Per this chance, we provide more details below regarding our application to the Treatment Allocation for Sepsis Patients, and have revised the contents per the reviewer’s suggestions. First, we construct our sepsis management environment using the MIMIC-III Clinical Database, from which we curate and preprocess 250,000 high-quality data points. The observed state space captures a wide range of clinical and physiological variables relevant to patient monitoring, including demographic information (such as age and gender), vital signs (heart rate, blood pressure, respiratory rate, oxygen saturation, and temperature), and neurological status. State transitions are modeled using a long short-term memory (LSTM) network to capture temporal dependencies, with additional noise to reflect real-world variability. The action space includes sepsis-critical interventions such as vasopressors and intravenous fluids, as well as medications like beta-blockers and diuretics that are typically used for managing comorbid conditions. To increase the complexity of the decision-making task, we further expand the action space with treatments that have an influence on body status but minimal relevance to sepsis, like antihistamine. The reward at each time step combines a SOFA-based component, which encourages clinical improvement, with a health-conditioned bonus that incorporates broader physiological indicators and treatment effects.
>
> The detailed list of states and actions can be found in **Table C.2 in the revised paper** with more elaborations about the environment design in **Appendix C.2 of the revised paper**. Our goal or motivation is to select and intervene on effective treatments for sepsis patients.
>
> Please refer to Section 5 and Appendix C.2 of the revised paper for more details.

---

> > ### Author Response · Authors · 2025-04-07
> >
> > > 3. Similarly to the previous point, the experiments could have been more convincing. While I liked Figure 4 as it shows basically how artificially adding redundant actions makes MuJoCo more difficult, I would have loved to see this figure for a domain that is more real. Are there every enough redundant actions that you can see a dramatic change like this? Where do these redundant actions arise from? Why couldn't we see a similar plot, for example, in the sepsis treatment domain?
> >
> > This is illustrated in our application to **the treatment allocation for sepsis patients**, where redundant actions may include treatments irrelevant to Sepsis but potentially effective for other conditions. Specifically, we consider a 30-dimensional action space that includes three types of actions: 1. Crucial treatments used in the intensive care units for sepsis patients like vasopressors and intravenous fluids, which are commonly administered in sepsis management to stabilize blood pressure and maintain fluid balance; 2. Other relevant but non-essential medications such as beta-blockers and diuretics, which—although not primarily intended for sepsis—may be prescribed to manage comorbid conditions like hypertension or fluid overload that often coexist with or complicate sepsis; and 3. Irrelevant or non-effective treatments like antihistamine. The type 2 and 3 treatments are considered less essential or potentially redundant in the context of sepsis management. The corresponding results, including learning curves and selected treatments, are summarized in **Figure 5 in Section 5 of the revised paper**, which demonstrates the effectiveness of the proposed method in identifying the useful treatments for sepsis patients.
> >
> > We have added more results and details on the construction of the Sepsis environment, which has also been updated to better reflect realistic and complex scenarios—see Section 5 and Appendix C.2 of the revised paper for details.
> >
> >
> > ***Responses to Requested Changes***
> >
> > > 1. There is no discussion about what "false discovery rate" (FDR) means. I don't know what this means, so I am having a hard time following the paper.
> >
> > Thanks for your comment. The False Discovery Rate (FDR) is the expected proportion of false positives (irrelevant variables) among all the variables that are declared as important (selected). Specifically, for a selected subset $\widehat{G}$ of the covariates constructed from data, the false discovery rate (FDR) associated with $\widehat{G}$ is formally defined as
> > $\operatorname{FDR}:=\mathbb{E} \lbrace |\widehat{G} \bigcap \mathcal{H}_0 |/ \max(1, |\widehat{G}|)\rbrace$, where $|\cdot|$ denotes the cardinality of a set. In variable selection, controlling FDR means we allow some tolerance for false positives, but we want that proportion to be small and controlled. We have added this to Section 2.3 of the revised paper.
> >
> > > 2. Section 2.3 requires much more detail. For example, (2.1) What is a "response vector"? Are these essentially labels for a supervised learning problem? Perhaps this would simply be cleared up again with a description of FDR / the setting you consider in this section.
> >
> > Thanks for your comment. The "response vector" refers to the outcomes used in supervised variable selection and can be interpreted as labels in a supervised learning context. In our RL setting, we adapt this concept by defining the response as a combination of immediate rewards and next states, which together serve as the supervisory signal for identifying influential actions.
> >
> > > (2.2) Equation (1) says $(\boldsymbol{X}, \widetilde{\boldsymbol{X}})_{\text {swap }(\Omega)} \stackrel{d}{=}(\boldsymbol{X}, \widetilde{\boldsymbol{X}})$, but in your definitions of these variables/operations, it is entirely unclear to me where there is any randomness.
> >
> > You can think of the data matrix as being sampled from an underlying data distribution. Under this view, the statement essentially means that after the swapping operation, the joint distribution of the covariates and their knockoffs remains unchanged — this is known as *exchangeability*. For a more detailed explanation, refer to Lemma 3.2 of [1].
> >
> > [1] Candes, Emmanuel, et al. "Panning for gold:‘model-X Knockoffs for high dimensional controlled variable selection." Journal of the Royal Statistical Society Series B: Statistical Methodology
> >
> >
> > > (2.3) How do you calculate feature importance scores?
> >
> > Feature importance scores can be calculated by any machine learning methods that treat original and knockoff features symmetrically, like Lasso, random forest, and neural networks, as mentioned in Appendix E. This is a design choice, and we choose Lasso in our experiments, as mentioned in Section 5.

---

> > > ### Author Response · Authors · 2025-04-07
> > >
> > > > (2.4) It says "set $W_j=f\left(Z_j, \widetilde{Z}_j\right)$ in such a way that higher values of $W_j$ indicate stronger evidence of $\boldsymbol{x}_j$ being influential covariate". Isn't this entirely dictated by the choice of $f$ ? How should one choose $f$ ? How do you know chich covariates are likely to be influential so as to design $f$ to satisfy this condition?
> > >
> > > In the knockoff-based variable selection, the feature importance function $f$ is critical for distinguishing influential variables from noise while controlling the false discovery rate (FDR). Although the choice of $f$ does not affect FDR control—as long as the knockoff construction is valid—a poorly chosen or underfitting $f$ can reduce statistical power, which refers to the method’s ability to correctly identify true signals. More expressive models, such as random forests or neural networks, can be used to define $f$ in order to capture nonlinear relationships between features and the response.
> > >
> > > > (2.5) Again, it would be very helpful if the final result (e.g. equation (2)) could be contextualized. What is accomplished with this filtering rule?
> > >
> > > In the knockoff variable selection, if $X_j$ is truly important, then it should show a stronger association with $Y$ than its knockoff $\tilde{X}_j$, so $Z_j>\tilde{Z}_j$, and $W_j>0$. If $X_j$ is a redundant variable, then it's interchangeable with its knockoff, and due to symmetry, it's just as likely for $\tilde{Z}_j>Z_j$ leading to $W_j<0$. So, large positive $W_j$ values suggest essential variables, while large negative $W_j$ values suggest the knockoff is more associated, and thus the original is likely redundant. Hence, in equation (2), the numerator term estimates the number of false positives - how often knockoffs "win" and appear essential. The term in the denominator counts the selected variables - how many original variables are essential. Then, the searches for the smallest threshold such that the estimated false discovery proportion is smaller than $\alpha$.
> > >
> > > We have substantially revised Section 2.3 to add more details to fully address all points raised by the reviewer.
> > >
> > >
> > > > 3. Under equation (3), it says "action selection reduces the dimensionality of the input action space, there by reducing bias in $Q$ function fitting". This warrants some more discussion, intuitively I would expect the opposite, since you are effectively restricting the hypothesis class over $Q$ functions to a smaller one (more bias).
> > >
> > > This is a great question. We agree that a common concern with feature reduction is the potential increase in bias due to a smaller hypothesis space, particularly when excluding variables that may influence the target. However, in our setting, *the redundant actions are conditionally independent of both the reward and the next state and thus provide no additional information*. These can be viewed as nuisance variables. If our method correctly identifies and excludes such actions, then, in principle, the reduction in hypothesis space does not introduce additional bias. In practice, including irrelevant features can sometimes increase bias by complicating the model-fitting process.
> > >
> > > Importantly, reducing the input dimensionality can help lower the bias and variance. Specifically, the error bound for the policy/value estimation using MLP is approximately at the order of $O(n^{-(\beta\over\beta+d)})$ for $d$ as the dimension, $n$ as the sample size, and $\beta$ as the Hölder smoothness parameter controlling the local smoothness, as discussed in the existing deep learning theories [2]. Hence, reducing the dimensionality of action space via the selection of minimal sufficient actions can improve the value function fitting.
> > >
> > > [2] Farrell, Max H., Tengyuan Liang, and Sanjog Misra. "Deep neural networks for estimation and inference." Econometrica
> > >
> > > We have incorporated the above clarification into the revised version of the paper.
> > >
> > >
> > > > 4. In equation (4), essentially you're suggesting to stop policy gradients through action dimensions that are redundant. Why is this the correct strategy? How does this allow the policy to ignore those dimensions (e.g., perform "action selection")? Why don't you additionally act according to $\mathbf{m} \odot \mathbf{a}$ instead of a sampled under the policy (or alternatively, is there any way to concretely use the learned mask to enforce that the policy acting in a manner consistent with the unmasked actions only)?
> > >
> > > Thank you for your insightful question. In Equation (4), we apply a hard mask to stop gradients through redundant action dimensions identified by our knockoff-based action selection. This masking encourages the policy to focus on informative dimensions, effectively performing action selection without changing the policy architecture or requiring reinitialization.

---

> > > > ### Author Response · Authors · 2025-04-07
> > > >
> > > > Since policy gradient methods update the policy by taking the gradient of the log-likelihood of the actions taken, by setting the likelihood of redundant action, its influence on policy update will be diminished. While the full action vector is still sampled, the masked gradient ensures learning is driven only by the selected actions.
> > > >
> > > > Regarding acting solely on selected actions: this would require modifying the policy network architecture, which we aim to avoid—especially in *adaptive settings where dynamic change points in the environment can shift the importance of different actions*. In such scenarios, repeated reinitialization would lead to a loss of knowledge acquired by the pre-trained model. In contrast, our method supports adaptive fine-tuning, enabling the model to retain and build upon its prior understanding.
> > > >
> > > >
> > > > > 5. In definition 4.1, there appears to be a missing word or something at the beginning of the definition. There appears to be a sentence starting with "property on the augmented data matrix […]".
> > > >
> > > > Thank you for pointing out this typo. We’ve included “Knockoff statistics $\mathbf{W}=\left(W_1, \ldots, W_p\right)^{\top}$” at the beginning of Definition 4.1, in the updated version.
> > > >
> > > >
> > > > > 6. In Theorem 4.3, what is "modified FDR"? What is $\alpha$?
> > > >
> > > > We’re grateful to elaborate more. For a selected subset $\widehat{G}$ of the covariates constructed from data and some pre-specified level $q\in (0,1)$, **the modified FDR (mFDR)** associated with $\widehat{G}$ is formally defined as $\operatorname{mFDR}:=\mathbb{E} \lbrace |\widehat{G} \bigcap \mathcal{H}_0 |/ (1/q+|\widehat{G}|) \rbrace$, where $|\cdot|$ denotes the cardinality of a set. The mFDR offers a less conservative measurement of FDR by adjusting the denominator, and hence controlling mFDR is a sufficient way of controlling FDR. And $\alpha$ is the target FDR level. The knockoff procedure controls the mFDR at a pre-specified level $\alpha$, ensuring that the approximated expected proportion of false positives among the selected variables does not exceed $\alpha$.
> > > >
> > > > We have added the above clarifications in Section 2.3 of the revised paper.
> > > >
> > > >
> > > > > 7. There is a notable lack of discussion regarding Theorem 4.3. Probably exacerbated by the difficulty I had following the background section, it is very hard to intuit the consequences of this theorem. It should be clearly explained how this bound applies to RL / policies, and why we want systems under which this upper bound is minimized.
> > > >
> > > > Thank you for your comments. Theorem 4.3 provides the theoretical guarantee for controlling the modified false discovery rate (i.e., mFDR) by our proposed method. Specifically, the upper bound consists of the target FDR level $\alpha$ and a small order term that vanishes as the sample size approaches infinity, by accounting for the cost of dependence. In reinforcement learning, this ensures the identification of a minimal sufficient set of actions with a controlled error tolerance, which is essential for efficient learning. Our method selects and only utilizes those relevant actions, avoiding irrelevant ones that could degrade policy performance and lead to unnecessary costs.
> > > >
> > > >
> > > > > 8. In the "Semi-synthetic MuJoCo Environments", what does it mean to "add extra actions to the raw action space"? What are these action dimensions? Are they just ignored by the simulator?
> > > >
> > > > Yes, in the semi-synthetic MuJoCo environments, those extra actions are just appended on the action space and ignored by the simulator. The experiments aim to show that even with such a “simple” design, the original deep RL method can easily fail and indicate the importance of action selection.
> > > >
> > > >
> > > > We sincerely hope that these clarifications and discussions satisfactorily respond to your comments. All these discussions and additional results are now part of the revised paper. We welcome any further questions or insights, and thank you once again for your vital contributions to our work!

---

> > > > > ### Comment · Reviewer_3oS8 · 2025-04-07
> > > > >
> > > > > Thanks to the authors for their responses and their edits to the manuscript.
> > > > >
> > > > > Admittedly there is a lot to digest here, and I have not spent nearly enough time on the revision to fully digest it yet. From what I've seen, I believe the revised draft makes some very substantial improvements with regard to clarity.
> > > > >
> > > > > ### Background on Knockoff Sampling
> > > > > The revised background on knockoff sampling is a huge improvement---especially due to the inclusion of explicit definitions of many of the relevant quantities such as FDR and mFDR. I still think certain parts should be further clarified. For example:
> > > > >
> > > > > 1. You specify the property that $(\pmb{X}, \widetilde{\pmb{X}})_{\mathrm{swap}(\Omega)} \overset{d}{=} (\pmb{X}, \widetilde{\pmb{X}})$. I'm assuming this equation is relating the joint distributions on both sides, and the swap operation happens for both components of that joint distribution simultaneously? Maybe I don't understand the swap operation, but from what I can tell, doesn't this also impose an assumption on $\pmb{X}$ (independently of the knockoff selection)? For instance, in order for this equation to hold, it seems that the distribution of $\pmb{X}$ itself would have to be invariant under permutations. For continuous action spaces, I suppose this can be forced by whitening, but what if you had discrete action space?
> > > > > 1. It still is not clear to me where these filters and feature importance functions come from. You say in the rebuttal that they can come from LASSO, random forests, or neural networks, but how are these models trained?
> > > > >
> > > > >
> > > > > ### Sepsis Experiment
> > > > > Thanks a lot for including more details about this experiment, this is much more interesting and relevant than the MuJoCo experiments, in my opinion.
> > > > >
> > > > > ### Knockoff Sampling in RL
> > > > > I still have some questions about the intricacies of applying knockoff sampling to RL.
> > > > >
> > > > > 1. I must've forgotten to ask in my original review, but what does it mean to sample a knockoff copy (as shown in the pseudocode)? The knockoff copy and the action preceding it are sampled from the same policy in the pseudocode, so it isn't clear to me what its purpose is.
> > > > > 1. Theorem 4.3 makes a seemingly very strong assumption on the data distribution, namely, that it is stationary. This seems overly strong for several reasons, particularly since you are aiming to do online RL. Firstly, over the course of online RL, the state distribution should be non-stationary since your policy is changing. And likewise, _especially_ since you're trying to learn which actions to ignore, the action distribution should absolutely be nonstationary as well, right? Second, which is related to my first point, is that it seems that you are essentially imposing this assumption in order to directly leverage the results about knockoff sampling in supervised learning to your RL setting. But in doing so, the assumption seemingly eliminates the characterizing properties of RL. Can you say anything about how (or if) the non-stationary nature of sequential decision-making under uncertainty influences knockoff sampling?

---

> > > > > > ### Author Response · Authors · 2025-04-08
> > > > > >
> > > > > > Thanks for your continued engagement and timely feedback. Below, we summarize your comments in quotes and provide our point-by-point responses.
> > > > > >
> > > > > > ***Background on Knockoff Sampling***
> > > > > >
> > > > > >
> > > > > > > 1. You specify the property that $(\boldsymbol{X}, \widetilde{\boldsymbol{X}})_{\text {swap }(\Omega)} \stackrel{d}{=}(\boldsymbol{X}, \widetilde{\boldsymbol{X}})$. I'm assuming this equation is relating the joint distributions on both sides, and the swap operation happens for both components of that joint distribution simultaneously? Maybe I don't understand the swap operation, but from what I can tell, doesn't this also impose an assumption on $\boldsymbol{X}$ (independently of the knockoff selection)? For instance, in order for this equation to hold, it seems that the distribution of $\boldsymbol{X}$ itself would have to be invariant under permutations. For continuous action spaces, I suppose this can be forced by whitening, but what if you had discrete action space?
> > > > > >
> > > > > >
> > > > > > We believe there is a misunderstanding regarding the swap operation. For example, if $\boldsymbol{X} \in \mathbb{R}^p$ is a $p$-dimensional variable and its knockoff copy $\widetilde{\boldsymbol{X}} \in \mathbb{R}^p$ is also $p$-dimensional, then specifying a swap set $\Omega = {1}$ means we exchange $X_1$ (the first dimension of $\boldsymbol{X}$) and $\widetilde{X}_1$ (the first dimension of $\widetilde{\boldsymbol{X}}$). If, after this swap, the joint distribution remains unchanged, we say that the knockoff features satisfy the **exchangeability** condition. *The detailed and more general definition of the swap operation can be found after Equation (1) on Page 5 in our paper*.
> > > > > >
> > > > > >
> > > > > > Second, the swap operation does not impose any additional assumption, while it defines the exchangeability condition that we rely on to **construct the knockoff variables** $\widetilde{\boldsymbol{X}}$. As we mentioned in the previous paragraph, the knockoffs must be constructed such that the joint distribution is invariant under coordinate-wise swaps between $\boldsymbol{X}$ and $\widetilde{\boldsymbol{X}}$.
> > > > > >
> > > > > >
> > > > > > Therefore, the swap is **NOT** performed between two dimensions of $\boldsymbol{X}$, but rather between corresponding coordinates of $\boldsymbol{X}$ and $\widetilde{\boldsymbol{X}}$. As a result, the distribution of $\boldsymbol{X}$ itself does not need to be permutation-invariant.
> > > > > >
> > > > > > > 2. It still is not clear to me where these filters and feature importance functions come from. You say in the rebuttal that they can come from LASSO, random forests, or neural networks, but how are these models trained?
> > > > > >
> > > > > > The feature importance score $Z_{i,j}$ for the $i-$th outcome (the reward or the next state) and the $j-$th action is computed by fitting this outcome based on all inputs using a machine learning method. Specifically, this process fits a predictive model $\hat{f}(x)$ to estimate a target variable $y$, where $y$ corresponds to either the reward or the next state, the feature vector $x$ comprises the current state, action, and a knockoff copy, and the function $f$ can be the function classes of LASSO, random forests, or neural networks. The importance score of a specific variable corresponds to its estimated coefficient in LASSO, its feature importance score in random forests, or its weight/gradient in neural networks. Separate models are constructed for each target outcome *as detailed in Algorithm 2*.
> > > > > >
> > > > > > ***Knockoff Sampling in RL***
> > > > > > > 1. I must've forgotten to ask in my original review, but what does it mean to sample a knockoff copy (as shown in the pseudocode)? The knockoff copy and the action preceding it are sampled from the same policy in the pseudocode, so it isn't clear to me what its purpose is.
> > > > > >
> > > > > > In short, a knockoff copy corresponds to the knockoff action variable, and sampling a knockoff copy means the specific construction of knockoff actions under our variable selection framework. In our setup, actions are drawn from a policy modeled as a diagonal Gaussian distribution, meaning each action dimension is independent given the state. Because of this independence structure, we can generate the knockoff action by independently resampling from the same policy distribution, conditioned on the same state. This construction satisfies the key requirement of exchangeability — the joint distribution of the original and knockoff actions remains invariant under swapping — since both are sampled from the same conditional distribution. Moreover, since the knockoff is generated conditionally on the state and independently of the outcome (reward or next state), it naturally satisfies the requirement of conditional independence from the target, making it a valid knockoff feature. With these properties in place, we can apply knockoff-based variable selection techniques directly on the state-action pair to identify important features while provably controlling the false discovery rate (FDR). *The detailed definition and construction can be found in Sections 2.3 and 3.1 in our paper*.

---

> > > > > > > ### Author Response · Authors · 2025-04-08
> > > > > > >
> > > > > > > > 2. Theorem 4.3 makes a seemingly very strong assumption on the data distribution, namely, that it is stationary. This seems overly strong for several reasons, particularly since you are aiming to do online RL. Firstly, over the course of online RL, the state distribution should be non-stationary since your policy is changing. And likewise, especially since you're trying to learn which actions to ignore, the action distribution should absolutely be nonstationary as well, right? Second, which is related to my first point, is that it seems that you are essentially imposing this assumption in order to directly leverage the results about knockoff sampling in supervised learning to your RL setting. But in doing so, the assumption seemingly eliminates the characterizing properties of RL. Can you say anything about how (or if) the non-stationary nature of sequential decision-making under uncertainty influences knockoff sampling?
> > > > > > >
> > > > > > > Stationarity is indeed a key assumption when adapting knockoff sampling to the reinforcement learning (RL) setting. Fortunately, even in online RL, this assumption often holds. For instance, popular deep RL algorithms such as PPO and SAC, typically **perform policy updates in batches**, where *the policy remains fixed for a number of interactions*. During each batch, the collected data can be regarded as stationary since actions are drawn from **the same policy**.
> > > > > > >
> > > > > > > Regarding your second point, we are not directly transferring knockoff sampling results from supervised learning to the RL setting. In fact, a critical distinction is that *the original knockoff framework assumes i.i.d. data, which does not hold in RL due to its sequential nature*. To address this, we incorporate exponential $\beta$-mixing, which explicitly **accounts for the temporal dependence among transition tuples**. This allows us to adapt knockoff sampling to sequential data while still ensuring valid inference under certain mixing conditions.
> > > > > > >
> > > > > > > While we argue that stationarity often holds in online RL, we acknowledge that handling fully non-stationary scenarios presents a significant theoretical challenge. Even in simpler settings like bandits, obtaining theoretical guarantees typically requires structural assumptions on the nature of non-stationarity—such as piecewise stationarity [1] or bounded variation in the underlying parameters [2]. Extending our framework to these cases is an interesting and important direction for future work.
> > > > > > > In practice, non-stationarity can often be mitigated by using data collected within short time windows. Common stabilization techniques—such as policy clipping or KL-regularization—limit the magnitude of policy updates, leading to relatively minor changes between updates. As a result, the data within each window can be treated as approximately stationary for practical purposes
> > > > > > >
> > > > > > > [1] Wu, Qingyun, Naveen Iyer, and Hongning Wang. "Learning contextual bandits in a non-stationary environment."
> > > > > > > [2] Russac, Yoan, Claire Vernade, and Olivier Cappé. "Weighted linear bandits for non-stationary environments.
> > > > > > >
> > > > > > >
> > > > > > > We sincerely hope that these clarifications and discussions satisfactorily respond to your comments. We welcome any further questions or insights, and thank you once again for your vital contributions to our work!

---

> > > > > > > > ### Comment · Reviewer_3oS8 · 2025-04-08
> > > > > > > >
> > > > > > > > ### Swap Operation
> > > > > > > > Thanks for the clarification, this makes sense and I understand now. Though, looking back at the paper, I still stand by that my interpretation is not unreasonable, so I think this should be made more precise in a revised draft.
> > > > > > > >
> > > > > > > > ### Feature Importance Scores
> > > > > > > > Again, interesting, and I think I understand now. I see how this comes into play in Algorithm 2 now, but it's still very implicit, you just say you train a model on tuples $(s_t, a_t, \tilde{a}_t, y_t)$, but this doesn't e.g. specify what the fitting objective is. A concrete discussion of how this works would be highly beneficial (though I believe now I understand from our discussion).
> > > > > > > >
> > > > > > > > ### Knockoff Copy
> > > > > > > > Thanks, I think I understand now. I last read and submitted the review for this paper in early December, so I'm in the midst of catching back up...
> > > > > > > >
> > > > > > > > ### Stationarity
> > > > > > > > Regarding stationarity, you say "Fortunately, even in online RL, this assumption often holds". Maybe I'm not understanding what you mean by "stationarity" then, because surely data in RL is rarely stationary. Even though some methods perform updates in batches which can make the data within a batch appear roughly stationary, _between_ updates I would expect things to change substantially enough that your estimates of the knockoff variables could be different, at which point, I don't know if your Theorem still holds (I'm not saying that it _doesn't_, just that I don't know, and I don't think you've proved it unless the data is actually stationary over the course of training). Can you, for example, guarantee that there is no oscillatory dynamics of the knockoff variables, or that the choice of knockoff variables will not impede policy optimization? I suppose you do not explicitly claim either of these, but implicitly the Theorem is quite a bit less meaningful if you lose policy optimization.
> > > > > > > >
> > > > > > > > Anyway, of course I appreciate that handling non-stationarity is an extreme challenge in RL, and probably this paper will not be the one to solve that (nor should it be). I guess my remaining question, to summarize the previous paragraph, is that despite non-stationarity being a challenge:
> > > > > > > > * We can at least hope to find local optima via e.g. policy gradient methods
> > > > > > > > * Is this still true when you do action masking according to your method?
> > > > > > > >
> > > > > > > > Regarding "exponential $\beta$-mixing"---is this defined anywhere in the text? I did see this mentioned, but I do not actually know what this means or how to interpret it the consequences of this assumption. I had assumed it was some mixing time assumption on the Markov chain induced by the learned policies, which I suppose would give you stationary data within an episode under a policy, but does not assert any stationarity across policy updates.

---

> > > > > > > > > ### Author Response · Authors · 2025-04-09
> > > > > > > > >
> > > > > > > > > Thanks for your continued engagement and timely feedback. Below, we summarize your comments in quotes and provide our point-by-point responses.
> > > > > > > > >
> > > > > > > > > **Swap Operation**
> > > > > > > > > >Thanks for the clarification, this makes sense and I understand now. Though, looking back at the paper, I still stand by that my interpretation is not unreasonable, so I think this should be made more precise in a revised draft.
> > > > > > > > >
> > > > > > > > > We have added the clarifications in Section 2.3 of the revised paper.
> > > > > > > > >
> > > > > > > > >
> > > > > > > > > **Feature Importance Scores**
> > > > > > > > > >Again, interesting, and I think I understand now. I see how this comes into play in Algorithm 2 now, but it's still very implicit, you just say you train a model on tuples $(s_t, a_t, \tilde{a}_t, y_t)$, but this doesn't e.g. specify what the fitting objective is. A concrete discussion of how this works would be highly beneficial (though I believe now I understand from our discussion).
> > > > > > > > >
> > > > > > > > > We have added the clarifications in Appendix G of the revised paper.
> > > > > > > > >
> > > > > > > > > **Stationarity**
> > > > > > > > >
> > > > > > > > > >Anyway, of course I appreciate that handling non-stationarity is an extreme challenge in RL, and probably this paper will not be the one to solve that (nor should it be). I guess my remaining question, to summarize the previous paragraph, is that despite non-stationarity being a challenge: We can at least hope to find local optima via e.g. policy gradient methods. Is this still true when you do action masking according to your method?
> > > > > > > > >
> > > > > > > > > We appreciate the reviewer’s thoughtful question. In our context, stationarity refers to the policy being fixed and the transition dynamics and reward function remaining unchanged. Hence, in the common batch update setting, the data collected between policy updates can be treated as stationary. Our method can accommodate different forms of non-stationarity.
> > > > > > > > >
> > > > > > > > > When non-stationarity arises solely due to policy updates—while the environment’s dynamics and rewards remain unchanged—our method can effectively handle this scenario by conducting knockoff variable selection within each batch. This approach boosts policy optimization by ensuring that action masking is informed by relevant, stable data. To enhance robustness, we can also extend our selection procedure to operate every k batch, leading to more stable and adaptive masking throughout training.
> > > > > > > > >
> > > > > > > > > In cases where non-stationarity stems from changes in the transition dynamics or reward function—which is rare in simulated environments like MuJoCo but plausible in real-world scenarios—the problem becomes more challenging. Nonetheless, our method remains effective under the realistic assumption of piecewise stationarity. By integrating change point detection, we can segment the data into stationary intervals and apply feature selection within each segment. This ensures that the selected action subset remains relevant and informative, thereby preserving and even enhancing policy optimization.
> > > > > > > > >
> > > > > > > > > In summary, our knockoff-based variable selection method can significantly enhance policy optimization under policy-induced non-stationarity. For environment-induced non-stationarity, our method remains effective as long as mild conditions—such as piecewise stationarity—are satisfied.
> > > > > > > > >
> > > > > > > > > We have added those discussions in Appendix H.
> > > > > > > > >
> > > > > > > > > >Regarding "exponential $\beta$-mixing"---is this defined anywhere in the text? I did see this mentioned, but I do not actually know what this means or how to interpret it the consequences of this assumption. I had assumed it was some mixing time assumption on the Markov chain induced by the learned policies, which I suppose would give you stationary data within an episode under a policy, but does not assert any stationarity across policy updates.
> > > > > > > > >
> > > > > > > > > Thank you for pointing this out. We have added the formal definition of the $\beta$-mixing coefficient and exponential $\beta$-mixing in Appendix I.1. The $\beta$-mixing coefficient quantifies the dependence between the distant past and future in a stochastic process. When a process is exponentially $\beta$-mixing, this dependence decays exponentially fast over time, meaning the process becomes increasingly independent as time steps increase. This condition is commonly used in theoretical analyses to approximate dependent data with i.i.d.-like behavior.
> > > > > > > > >
> > > > > > > > > In our setting, when data is collected within a batch under a fixed policy, the process is stationary. Under the exponential $\beta$-mixing assumption, we can apply sample splitting (as described in Algorithm 2) to ensure that the transition tuples in each split are approximately independent. This allows for the construction of valid knockoff statistics for variable selection. Moreover, Theorem 4.3 explicitly accounts for this approximation error in the derived error bound.
> > > > > > > > >
> > > > > > > > > We sincerely hope that these clarifications and discussions satisfactorily respond to your comments. Thank you once again for your vital contributions to our work!

---

### Review · Reviewer_1D57 · 2024-12-11

**Summary Of Contributions:**

This paper introduced a new framework for RL with high dimensional action spaces by combining several techniques from other research areas like model-X knockoff and sample splitting. The paper is in general well-written and sufficiently motivated, with experimental results backing the proposed action selection method.

**Audience:**

Yes

**Claims And Evidence:**

Yes

**Requested Changes:**

Please refer to the my two points above.

**Strengths And Weaknesses:**

Handling high dimensional action spaces is a less studied but potentially important direction. I can see one example is the treatment allocation environment given by the authors where the agent is required to shrink the enormous amount of possible treatment combinations to a meaningful subset. The paper is well-organized in detailing the relevant techniques borrowed from related literature and how these tools should be properly handled when dealing with non-iid reinforcement learning data.

While the paper is interesting, I believe there is room for making the paper more convincing. My suggestions are mainly the following two points:
 - **methodologically**, Eq. 4 looks quite restricting by transforming the log-likelihood into a binary vector.  The authors' claim that
> This masking of the log probability helps mitigate the likelihood of encountering extremely high entropy values, thereby facilitating a more stable and efficient training process.

is not convincing and seems to bias towards only the good side. When we take the gradient of log-likelihood for updating the policy, the information carried by $\nabla \log \pi(a|s)$ is lost by this binary vector. Careful expositions are needed here to clearly state why the authors did not use $\odot$ like Eq. 3 but rather dot product. More importantly, the authors should explain, or at least conjecture why the current design leads to good performance. On this matter, perhaps the authors can link it to the sparse filtering log-likelihood update, see refs [1,2]. Specifically, we can consider the following more general policy learning objective $\mathbb{E}_{s,a \sim \mathcal{D}}[ w(s,a) \log \pi(a|s) ] $.
Sparse filtering log-likelihood update designates $w(s,a)$ to be 0 for certain actions (in this context, the redundant actions ).

 - **empirically**, I think the paper would benefit greatly from emphasizing more on the simulated treatment environment that runs on the MIMIC-III dataset. Artificial action dimensions for MuJoCo serve as a proof-of-concept example and are less interesting. I would encourage the authors to first show results on this environment, and provide more details including how states, actions and rewards are designed; how high dimensional action spaces emerge naturally from the combination of possible treatments; and learning curves.

References: \
[1] Offline RL with no ood actions: in-sample learning via implicit value regularization\
[2] Offline RL via Tsallis Regularization

---

> ### Author Response · Authors · 2025-04-07
>
> We are very grateful for your valuable comments and insightful feedback! In response to your comments, we make the following clarifications and additional revisions to the paper. Below, we summarize your comments in quotes and provide our point-by-point responses. Please refer to the latest submission for the revised paper, taking all your suggestions using **blue** color.
>
> > Methodologically, Eq. 4 looks quite restricting by transforming the log-likelihood into a binary vector. The authors' claim that “This masking of the log probability helps mitigate the likelihood of encountering extremely high entropy values, thereby facilitating a more stable and efficient training process” is not convincing and seems to bias towards only the good side. When we take the gradient of log-likelihood for updating the policy, the information carried by $\nabla$ log $\pi(a \mid s)$ is lost by this binary vector. Careful expositions are needed here to clearly state why the authors did not use $\odot$ like Eq. 3 but rather dot product.
>
> Thank you for your insightful comments. The reason we utilize the dot product in this context stems from the properties of multivariate Gaussian policies, where actions are independent. The log likelihood of an action can be expressed as $\log \pi_\theta^m(\mathbf{a} \mid \mathbf{s}) = \prod_{i=1}^p \log \pi_\theta(a_i \mid \mathbf{s})$. To address situations where certain action dimensions may not be beneficial, we can selectively filter the log-likelihood using $\mathbf{m} \cdot (\log \pi_\theta(a_1 \mid \mathbf{s}), \ldots, \log \pi_\theta(a_p \mid \mathbf{s})) = \prod_{i=1}^p m_i \log \pi_\theta(a_i \mid \mathbf{s})$. This method *effectively removes unnecessary actions during the training of the policy when actions are independent*. Our approach can also be adapted for correlated actions by applying a mask to the correlation matrix, with further details provided in Appendix E.
>
> > More importantly, the authors should explain, or at least conjecture, why the current design leads to good performance. On this matter, perhaps the authors can link it to the sparse filtering log-likelihood update (see refs [1,2]. Specifically, we can consider the following more general policy learning objective $\mathbb{E}_{s, a \sim \mathcal{D}}[w(s, a) \log \pi(a \mid s)]$. Sparse filtering log-likelihood update designates $w(s, a)$ to be 0 for certain actions (in this context, the redundant actions ).
>
> We appreciate this opportunity to elaborate on the reason for our good performance and alternative choices. First, the performance gain can come from estimation error with high-dimensional actions. The error bound for the policy/value estimation using MLP is approximately at the order of $O(n^{-(\beta\over\beta+d)})$ for $d$ as the dimension, $n$ as the sample size, and $\beta$ as the Hölder smoothness parameter controlling the local smoothness, as discussed in the existing deep learning theories [1]. Hence, reducing the dimensionality of action space via the selection of minimal sufficient actions can improve the value function fitting. Second, our method primarily focuses on pruning redundant action dimensions, improving model efficiency by selectively removing less informative actions. While the sparse filtering of log-likelihood updates, mentioned by the reviewer, goes beyond simple dimension reduction—by setting the joint probability of actions to zero when their associated weights are zero, thereby inducing strong sparsity—our approach introduces sparsity at the individual dimension level. Specifically, we set the log-likelihood to zero for certain action dimensions without collapsing the entire joint distribution.
>
> [1] Farrell, Max H., Tengyuan Liang, and Sanjog Misra. "Deep neural networks for estimation and inference." Econometrica

---

> ### Author Response · Authors · 2025-04-07
>
> > Empirically, I think the paper would benefit greatly from emphasizing more on the simulated treatment environment that runs on the MIMIC-III dataset. Artificial action dimensions for MuJoCo serve as a proof-of-concept example and are less interesting. I would encourage the authors to first show results on this environment, and provide more details including how states, actions and rewards are designed; how high dimensional action spaces emerge naturally from the combination of possible treatments; and learning curves.
>
> We are grateful for your valuable comments. We provide more details below and have revised the contents per the reviewer’s suggestions. First, we construct our sepsis management environment using the MIMIC-III Clinical Database, from which we curate and preprocess 250,000 high-quality data points. The observed state space captures a wide range of clinical and physiological variables relevant to patient monitoring, including demographic information (such as age and gender), vital signs (heart rate, blood pressure, respiratory rate, oxygen saturation, and temperature), and neurological status. State transitions are modeled using a long short-term memory (LSTM) network to capture temporal dependencies, with additional noise to reflect real-world variability. The action space includes sepsis-critical interventions such as vasopressors and intravenous fluids, as well as medications like beta-blockers and diuretics that are typically used for managing comorbid conditions. To increase the complexity of the decision-making task, we further expand the action space with treatments that have an influence on body status but minimal relevance to sepsis, like antihistamine. The reward at each time step combines a SOFA-based component, which encourages clinical improvement, with a health-conditioned bonus that incorporates broader physiological indicators and treatment effects.
>
> The detailed list of states and actions can be found in **Table C.2 in the revised paper** with more elaborations about the environment design in **Appendix C.2 of the revised paper**. The corresponding results, including learning curves and selected treatments, are summarized in **Figure 5 in Section 5 of the revised paper**, which demonstrates the effectiveness of the proposed method in identifying the useful treatments for sepsis patients.
>
>
> We sincerely hope that these clarifications and discussions satisfactorily respond to your comments. All these discussions and additional results are now part of the revised paper. We welcome any further questions or insights, and thank you once again for your vital contributions to our work!

---

### Review · Reviewer_wPAB · 2025-03-23

**Summary Of Contributions:**

This paper focuses on RL in the setting of high-dimension actions, where the straightforward application of existing exploration methods requires significant computation. The existing works reduce the action space through eliminating action space redundancy using domain expertise or require action spaces to be discrete and suffer from high computational cost. The paper proposes a new data-driven strategy for reducing the number of action selections through knockoff sampling. The paper provides theoretical guarantees and empirical validation of the approach.

**Audience:**

Yes

**Claims And Evidence:**

Yes

**Requested Changes:**

Please refer to the comments and questions in the weakness above. Additional questions:

- Could you please explain the challenges involved in extending the knockoff method of Ma et al. 2023 from offline and high-dimensional state space setting to online and high-dimensional action space setting, from both theoretical analysis and algorithm design standpoints?
- Compared to the works Zahavy et al. 2018 and Zhong et al. 2024, could you please explain how your approach is superior or different, considering the naive extension of these works by discretizing the action space?
- Is it possible to extend the definition of minimal action set to only hold approximately? In that case, how are the theoretical results impacted?

**Strengths And Weaknesses:**

Strengths
- This is an import challenge in RL to develop practical techniques that can efficiency handle high-dimensional actions.
- The paper is very well-written and clear.
- The paper makes multiple contributions, including formulating sufficient and minimial sufficient actions sets and a knockoff sampling techniques for online setting and continuous action spaces, which is practical.
- It’s nice to see experiments on both (though semi-synthetic) MuJoCo tasks and on clinical data. It’s interesting that even in some cases, the KS approach nearly matches PPO with true actions.

Weaknesses
- In experiments, high-dimensional action space is artificially created by adding extra actions, which may not capture the complexities of real scenarios.
- MoJuCo experiments only compare KS with True Actions and All actions, and comparison with other methods are not presented. For example, how does a simple low-rank approximation of actions perform? What about the approach of Zahavy et al. 2018 by discreteizing the action space? Additionally, are there any action limitation benefits to bonus-based exploration methods that leverage function approximation such as RND?
- Theoretical contribution is limited and appear to be straightforward extension of Ma et al. 2023. It would be good if the authors clarify differences with Ma et al. 2023.

---

> ### Author Response · Authors · 2025-04-07
>
> We are very grateful for your valuable comments and insightful feedback! In response to your comments, we made the following clarifications and additional revisions to the paper. Below, we summarize your comments in quotes and provide our point-by-point responses. Please refer to the latest submission for the revised paper, taking all your suggestions using **blue** color.
>
> ***Responses to Weaknesses***
>
> > 1. In experiments, high-dimensional action space is artificially created by adding extra actions, which may not capture the complexities of real scenarios.
>
> Thank you for your valuable comments. First, we would like to clarify that the high-dimensional action space with redundant actions constructed in our experiments aligns with our setting and serves as a foundational basis for exploring the necessity of action selection when there is redundancy. More specifically, the true actions affecting the reward of interest in MoJuCo are "sufficient action set" defined in Section 2.1, and redundant high-dimensional are created by adding extra actions. In addition, our experiments can be further extended to more complex situations where the redundant actions influence redundant states but do not cumulatively contribute or affect the reward in the long run, as mentioned in Section 2.2.
>
> Second, we do acknowledge the importance of capturing the complexities of real-world scenarios. Per your suggestion, in our application to **the treatment allocation for sepsis patients**, we consider a 30-dimensional action space that includes three types of actions, the crucial treatments used in the intensive care units for sepsis patients like vasopressor, the relevant but non-essential treatments such as beta-blockers and diuretics that influence the transition dynamics but may be redundant or less critical in the context of sepsis management, and irrelevant or non-effective treatments like antihistamine. The detailed list of states and actions can be found in **Table C.2 in the revised paper**. The corresponding results in **Figure 5 of the revised paper** demonstrate the effectiveness of the proposed method in identifying the useful treatments for sepsis patients.
>
> We’ve provided more illustrations and discussions regarding complex real applications in Section 5 and Appendix C.2 of our paper.
>
>
> > 2. MoJuCo experiments only compare KS with True Actions and All actions, and comparison with other methods are not presented. For example, how does a simple low-rank approximation of actions perform? What about the approach of Zahavy et al. 2018 by discreteizing the action space? Additionally, are there any action limitation benefits to bonus-based exploration methods that leverage function approximation such as RND?
>
> We thank the reviewer for your insightful comments and for pointing out these methods. First, a simple low-rank approximation of actions is close to the state abstraction methods (Misra et al., 2020; Pavse & Hanna, 2023) to learn a mapping from the original state space to a much smaller abstract space. However, as we discussed in our related work section, it is **hard to utilize such low-rank or abstraction-based methods to implement and interpret transformed actions**. Second, Zahavy et al. (2018) proposed an action elimination method that relies on **explicit elimination signals provided by the environment**—external feedback indicating whether an action is invalid in a given state. In contrast, our method does not depend on any such explicit signal; instead, it implicitly identifies essential actions through learned representations. In addition, the approach by Zahavy et al. (2018) does not directly target the issue of high-dimensional and continuous actions while indeed introducing more dimensionality by discretizing the continuous action space. Third, we agree that exploring potential action limitation benefits of bonus-based methods such as RND, particularly under function approximation, is an important direction—especially for **improving safety or efficiency in exploration**. While this aspect is not the focus of our current work, we acknowledge its relevance and consider it a valuable direction for future research.

---

> ### Author Response · Authors · 2025-04-07
>
> We would like to emphasize that in our experiment of the Treatment Allocation for Sepsis Patients, we DID include two additional baselines: Lattice (Chiappa et al., 2024) and gSDE (Raffin et al., 2022), for **comparison**. In the semi-synthetic MuJoCo environments, we only consider True Actions and All Actions to conduct the control experiment and demonstrate the importance and effectiveness of action selection. These experiments show that standard deep RL methods for continuous control can easily fail without a deliberate selection procedure. Our method acts as a straightforward plug-in solution to these challenges, circumventing the need for intricate designs and extensive hyperparameter tuning. This simplicity offers a stark contrast to more complex baseline methods, such as those described by Zahavy et al. (2018).
>
> We’ve added the above discussions to the related work in Section 1.1 and Appendix Section A of the revised paper.
>
>
> > 3. Theoretical contribution is limited and appear to be straightforward extension of Ma et al. 2023. It would be good if the authors clarify differences with Ma et al. 2023.
>
> We appreciate this opportunity to clarify our theoretical contribution. Our theoretical results can be seen as a **generalization** of the work by Ma et al. (2023), which focused on providing guarantees for state selection. In contrast, our research extends these guarantees to action selection. A significant distinction between our approaches lies in **the construction of the exchangeability condition**. Ma et al. (2023) imposed a stronger assumption, requiring a normal distribution of the variables and the use of second-order machinery to construct knockoff features. On the other hand, our method for building the exchangeability condition leverages the inherent structure of our problem, where actions can be directly resampled from the policy. By reducing assumptions, we enhance the applicability and realism of our approach. Thus, our theoretical contribution is not merely a straightforward extension; it is a tailored adaptation to this specific setting for high-dimensional action space, significantly boosting its relevance and practical utility.
>
>
> ***Responses to Requested Changes***
>
> > 4. Could you please explain the challenges involved in extending the knockoff method of Ma et al. 2023 from offline and high-dimensional state space setting to online and high-dimensional action space setting, from both theoretical analysis and algorithm design standpoints?
>
> Thank you for your insightful question. Here, we summarize the challenges in both theoretical analysis and algorithm design perspectives as follows.
>
> **Theoretical Perspective:** Theoretically, a significant challenge arises from *the strong data-generating assumptions in Ma et al. (2023), which we found to be overly restrictive for online learning scenarios*. Our approach relaxes these assumptions by leveraging the inherent properties of action selection in an online context, as mentioned in our response to 'Weakness #3', thus mitigating the need for stringent assumptions and broadening the applicability of our method in more realistic settings.
>
> **Algorithmic Perspective:** In the offline setting described by Ma et al. (2023), the algorithm performs variable selection as a separate stage before applying a deep reinforcement learning algorithm. However, *this two-stage approach poses challenges in online settings, where dynamic environments can introduce change points that require frequent updates to the sufficient sets*. To address this, our approach seamlessly integrates action selection into online deep RL methods through a masking mechanism, enabling dynamic adjustment of the sufficient sets in response to environmental changes and ensuring continuous, context-relevant learning and decision-making. We validate this integration with advanced deep RL algorithms such as Proximal Policy Optimization (PPO) and Soft Actor-Critic (SAC), which were not addressed in Ma et al. (2023).
>
> We’ve added the above discussions to the related work in Section 1.1 and Appendix Section A of the revised paper.

---

> ### Author Response · Authors · 2025-04-07
>
> > 5. Compared to the works Zahavy et al. 2018 and Zhong et al. 2024, could you please explain how your approach is superior or different, considering the naive extension of these works by discretizing the action space?
>
> We thank the reviewer for your insightful comments. First, Zahavy et al. (2018) proposed an action elimination method that relies on explicit elimination signals provided by the environment—external feedback indicating whether an action is invalid in a given state. In contrast, our method does not depend on any such explicit signal; instead, it implicitly identifies essential actions through learned representations. Second, Zhong et al. (2024) introduced a method that first fits an inverse dynamics model of the environment and then uses it to construct an action mask. However, fitting such a probabilistic model is particularly challenging in high-dimensional state spaces. Moreover, this mask-learning process is conducted entirely prior to policy training and is fully decoupled from it, suffering from the same limitations identified in Ma et al. (2023)—namely, a lack of adaptability during online learning.
>
> Extending these action elimination approaches to *continuous action spaces* typically involves discretizing the action space, which introduces two key challenges:
>
> **1. Potential Loss of Precision:** Discretization inevitably introduces approximation errors, as continuous actions are rounded to the nearest discrete bin. While increasing the discretization resolution can reduce this loss of precision, it leads to an explosion in the number of possible actions, making learning inefficient or even intractable. This reflects a fundamental trade-off between approximation fidelity and the curse of dimensionality.
>
> **2. Difficulty in Tuning the Number of Bins:** The performance of discretization-based methods is highly sensitive to the number of bins used. Different bin choices can lead to significantly different performance outcomes, and no principled theoretical framework exists for selecting the optimal binning strategy [1]. The problem becomes even more pronounced when each action dimension exhibits different characteristics—using a uniform bin size across dimensions often proves suboptimal. As the dimensionality increases, tuning these bins individually becomes increasingly complex and computationally expensive, limiting the scalability of such approaches.
>
> In contrast, our method operates directly in the continuous action space, making it inherently more scalable and precise. By preserving the native granularity of actions, our approach enhances both the effectiveness and efficiency of the policy learning process.
>
> We’ve added the above discussions to the related work in Section 1.1 and Appendix Section A of the revised paper.
>
>
> > 6. Is it possible to extend the definition of minimal action set to only hold approximately? In that case, how are the theoretical results impacted?
>
> This is an excellent question. Actually, since our framework is based on Knockoff, it indeed implies an approximation of the minimal action set of variables. The Knockoff offers a nuanced approach to variable selection that effectively balances the identification of influential variables with the control of false discoveries, particularly in complex, high-dimensional data settings. It controls the false discovery rate (FDR), focusing on limiting the number of erroneously selected variables. It selects variables that are most likely to influence the response while ensuring a statistically controlled proportion of false positives. Therefore, by increasing the false discovery rate ($\alpha$), our method can control the tolerance level—make the size of the selected action set less stringent and regard it as an optimistic approximation of a minimal sufficient set.
>
>
> [1] Tang, Yunhao, and Shipra Agrawal. "Discretizing continuous action space for on-policy optimization." Proceedings of the aaai conference on artificial intelligence. Vol. 34. No. 04. 2020.
>
>
> We sincerely hope that these clarifications and discussions satisfactorily respond to your comments. All these discussions and additional results are now part of the revised paper. We welcome any further questions or insights, and thank you once again for your vital contributions to our work!

---

### Decision · Action_Editor_3Kmf · 2025-06-11

**Recommendation:** Accept with minor revision

**Additional Comments:**

This paper presents a novel approach to the challenge of RL with high-dimensional action spaces by leveraging knockoff sampling for action selection. The authors provide theoretical guarantees and empirical validation. Reviewers recognized that the problem is practical and the solution is novel. While the reviewers have raised several concerns, including unclear presentation, lack of comparison to related works, and strong assumptions, the paper is greatly improved during revision. Reviewers recognized the substantial improvements and all leaned to accept the paper. One minor suggestion for camera-ready is to clarify $\beta$-mixing in the paper as shown in the discussion with Reviewer 3oS8: please consider moving the definition to the main paper and discussing the technical motivation to adopt it.

**Audience:**

Yes

**Audience Explanation:**

Yes.

**Claims And Evidence:**

Yes

**Claims Explanation:**

Yes. The claims of the paper are supported by the theoretical results and empirical validation.